# Understanding Generalization in Quantum Machine Learning with Margins

**Tak Hur** [1]  **Daniel K. Park** [1] [2]

## Abstract

Understanding and improving generalization capabilities is crucial for both classical and quantum machine learning (QML). Recent studies have revealed shortcomings in current generalization theories, particularly those relying on uniform bounds, across both classical and quantum settings. In this work, we present a margin-based generalization bound for QML models, providing a more reliable framework for evaluating generalization. Our experimental studies on the quantum phase recognition dataset demonstrate that margin-based metrics are strong predictors of generalization performance, outperforming traditional metrics like parameter count. By connecting this margin-based metric to quantum information theory, we demonstrate how to enhance the generalization performance of QML through a classical-quantum hybrid approach when applied to classical data.

## 1. Introduction

Quantum machine learning (QML) presents exciting opportunities to expand the horizons of machine learning beyond classical approaches. Generalization—the ability to learn from examples and make accurate predictions on unseen data—is a core component of intelligence and a critical factor that quantifies the effectiveness of machine learning models in real-world applications. Thus, fundamental challenge in QML, as in classical machine learning, is to understand, characterize, and optimize generalization. Generalization in QML has been studied through various factors, such as the number of parameters (Caro et al., 2022), effective dimension (Abbas et al., 2021a), quantum resource theory (Bu et al., 2021; 2022; 2023), and quantum information-theoretic

quantities (Banchi et al., 2021; Caro et al., 2024). Despite these efforts, existing methods have shown limitations in fully capturing the behavior of QML models.

The current understanding of generalization in QML predominantly relies on uniform bounds, which hold uniformly across all hypotheses within a given class. However, in classical deep learning, uniform bounds have faced significant criticism for their lack of practical relevance (Zhang et al., 2021; Nagarajan & Kolter, 2019; Dziugaite & Roy, 2017). In particular, Zhang et al. (2021) demonstrated that modern deep learning models can easily overfit random labels. Since uniform bounds apply equally to all hypotheses in the function family, this result indicates that these bounds fail to distinguish between models that generalize well and those that merely memorize data, highlighting their vacuity in deep learning. Building on this observation, Gil-Fuster et al. (2024) explored the use of parameterized quantum circuits, also known as Quantum Neural Networks (QNNs), on a benchmark quantum dataset with randomized labels. Despite the small number of qubits, the QNNs were able to overfit the random labels, indicating that uniform bounds are equally ineffective in the QML framework.

In classical deep learning, the looseness of uniform bounds has prompted many researchers to shift their focus toward explanatory tools, such as exploring their correlation with observed generalization. Notably, Bartlett et al. (2017) proposed a margin-based generalization bound for deep networks and demonstrated that spectrally normalized margin distribution is strongly correlated with generalization performance. Subsequent research has consistently found that margin-based metrics are reliable predictors of generalization performance (Neyshabur et al., 2018; Jiang et al., 2018; Dziugaite et al., 2020).

In this work, we demonstrate that margins are strong predictors of generalization performance within the QML framework as well. Our goal is to shift the focus away from the commonly emphasized parameter count, highlighting margins as a more effective tool for evaluating and controlling generalization. Moreover, this margin-based perspective provides a systematic pathway to improving the generalization of QML models when applied to classical data, as it reveals that maximizing the separability of classes embedded in the quantum feature space is key to optimizing

[1]Department of Statistics and Data Science, Yonsei University, Seoul, Republic of Korea [2]Department of Applied Statistics, Yonsei University, Seoul, Republic of Korea. Correspondence to: Tak Hur <takh0404@yonsei.ac.kr>, Daniel K. Park <dkd.park@yonsei.ac.kr>.

*Proceedings of the $42^{nd}$ International Conference on Machine Learning*, Vancouver, Canada. PMLR 267, 2025. Copyright 2025 by the author(s).

performance.

The central contributions of this work are:

- We establish margin-based generalization bound for multiclass classification with QNNs by adapting the approach of Bartlett et al. (2017), originally developed for classical neural networks, to the quantum domain. This involves interpreting quantum measurements as Lipschitz continuous nonlinear activations and extending matrix covering techniques to complex-valued spaces. The unitary constraints of quantum circuits and normalization of quantum states simplify the complexity of the original framework, enabling us to assess generalization performance through the margin normalized by the spectral norms of the measurement operators. This margin-based bound yields tighter upper bound when the hypothesis achieves larger margins on the sample set, addressing the limitations of uniform bounds in capturing meaningful generalization behavior.

- We experimentally demonstrate a strong correlation between generalization and margin distribution, even in scenarios with random labels, where traditional metrics like parameter count are ineffective. Furthermore, we compare margin-based metrics (e.g., lower quartile, median, and mean) against parameter-based metrics, consistently finding margins to be more reliable predictors of generalization in QNNs.

- We further establish a connection between margins and quantum state discrimination, a core concept in quantum information theory. This insight underscores the pivotal role of quantum embeddings in both optimization and generalization, providing valuable guidance for designing effective QML models when applied to classical data. We demonstrate that Neural Quantum Embedding (NQE) (Hur et al., 2024), a classical-quantum hybrid approach optimized for high data distinguishability, yields large margins, thereby enhances generalization performance.

## 2. Related Works

The concept of margin was popularized in machine learning by the development of support vector machines (Cortes & Vapnik, 1995). Following this, the theoretical connections between large margins and generalization bounds were explored for model classes such as two-layer neural networks (Bartlett, 1996) and linear classifiers (Shawe-Taylor et al., 1998), establishing margin analysis as a valuable tool for understanding generalization. In the context of modern deep learning, spectrally normalized margin bounds were introduced, derived from perspectives including covering number arguments (Bartlett et al., 2017) and PAC-Bayesian

theory (Neyshabur et al., 2018). More recently, Jiang et al. (2018) have underscored the importance of analyzing the margin distribution across training samples for accurately predicting the generalization gap.

Concurrently, efforts to understand generalization in QML have often involved extending established techniques from classical learning theory. These include metrics such as parameter counts (Caro et al., 2022), effective model dimensions (Abbas et al., 2021b), and quantum information-theoretic measures (Caro et al., 2024). However, a dedicated extension of margin-based generalization analysis to the QML domain has remained a significant gap. This paper directly addresses this by developing a margin-based framework for QML models, aiming to provide more effective predictions of their generalization capabilities.

The process of data embedding is another critical aspect when applying QML models to classical data, with demonstrable impacts on approximation error (Schuld et al., 2021), optimization landscapes (Holmes et al., 2022), and on generalization error (Caro et al., 2021). Specifically, Caro et al. (2021) investigated the role of quantum embedding on generalization utilizing fat-shattering arguments. In Section 5, we introduce a novel approach to investigating the influence of data encoding on generalization by leveraging our margin-based framework. We establish a connection between the margin mean and the trace distance, a key measure of quantum state distinguishability. This perspective not only contributes to the theoretical understanding of how data embeddings affect generalization but also offers a practical guide for enhancing the performance of QML models.

## 3. Margin Bound for Quantum Neural Networks

### 3.1. Multiclass Classification with Quantum Neural Networks

Consider an unknown joint probability distribution $\mathcal{D}$ governing the quantum state $\rho$ and its corresponding label $y$, where $\rho \in \mathbb{C}^{N \times N}$ and $y \in [k]$, representing an $n$-qubit ($n = \log_2 N$), $k$-class classification task. QNNs employ a parameterized quantum circuit $U(\theta)$, along with a set of positive operator-valued measurements (POVMs) $\{E_i\}_{i=1}^k$, to perform classification. Here, $U(\theta)$ is a unitary operator parameterized by $\theta$, satisfying $UU^\dagger = U^\dagger U = I$, where $U^\dagger$ is the conjugate transpose of $U$. POVMs are a generalization of projective measurements in quantum mechanics, where $\{E_i\}_{i=1}^k$ represents a set of positive semidefinite operators that sum to the identity. Specifically, the QNN maps a quantum state to a $k$-dimensional vector, $h_\theta(\rho) = \{\text{Tr}(U(\theta)\rho U^\dagger(\theta)E_i)\}_{i=1}^k$, where each element represents the probability of the state being assigned to a particular class.

Given $m$ independent and identically distributed (i.i.d.) samples $S = \{(\rho_i, y_i)\}_{i=1}^m$, the goal is to find a hypothesis $h^*$ (or optimal parameters $\theta^*$) that minimizes the true error, $R(h^*) = \mathbb{E}_{(\rho,y)\sim\mathcal{D}}[\mathbb{1}(\arg\max_j h^*(\rho)_j \neq y)]$, where $\mathbb{1}(x) = 1$ if $x$ is true, and 0 otherwise. Since the true distribution $\mathcal{D}$ is unknown, we instead seek a hypothesis $h$ (from a hypothesis class $\mathcal{H}$) with small empirical risk, $\hat{R}(h) = m^{-1}\sum_{i=1}^m \mathbb{1}(\arg\max_j h(\rho_i)_j \neq y_i)$.

For a given hypothesis $h$, the generalization gap is defined as the difference between the true and empirical risks, $g(h) = R(h) - \hat{R}(h)$. A common approach to understanding generalization is to upper bound $g(h)$ using a complexity measure that depends on the hypothesis class, $\mathcal{H} = \{\rho \mapsto \{\mathrm{Tr}(U\rho U^\dagger E_i)\}_{i=1}^k : U \in \mathbb{U}_Q\}$, where $\mathbb{U}_Q$ denotes the space of unitaries accessible by QNNs. The complexity of $\mathcal{H}$ depends on hyperparameters such as the quantum circuit architecture, the choice of unitary ansatz, the number of unitary layers, and the selection of POVMs.

### 3.2. Margin Generalization Bound

The concept of margin has been extensively explored since the early days of machine learning, offering theoretical foundations for support vector machines (Cortes & Vapnik, 1995). Recently, margin is adopted to understand generalization in deep learning (Bartlett et al., 2017; Neyshabur et al., 2018).

Margin generalization utilizes the ramp loss function $l_\gamma : \mathbb{R} \mapsto \mathbb{R}^+$,

$$l_\gamma(x) = \begin{cases} 0, & \text{if } x > \gamma \\ 1 - x/\gamma, & \text{if } 0 \leq x \leq \gamma \\ 1, & \text{if } x < 0. \end{cases}$$

For multiclass tasks, we define the margin operator $\mathcal{M}$ as $\mathcal{M}(v, y) = v_y - \max_{i \neq y} v_i$, measuring the gap between the correct label's probability and the highest competing label.

To derive margin bound for QNNs, we utilize *Rademacher complexity* on the function class $\mathcal{F}_\gamma = \{(\rho, y) \mapsto l_\gamma(\mathcal{M}(h(\rho), y)) : h \in \mathcal{H}\}$ (Mohri et al., 2018). Let $\sigma_i$ be i.i.d. Rademacher random variables, each taking values $\pm 1$ with equal probability. For a sample $S = \{(\rho_i, y_i)\}_{i=1}^m$, the sample Rademacher complexity is defined as $\mathfrak{R}((\mathcal{F}_\gamma)_{|S}) = m^{-1}\mathbb{E}[\sup_{f \in \mathcal{F}_\gamma} \sum_{i=1}^m \sigma_i f((\rho_i, y_i))]$. Then, for any $\delta > 0$ and $\gamma > 0$, with probability at least $1 - \delta$ over the random draw of an i.i.d. sample $S$ of size $m$, the following inequality holds for all $h \in \mathcal{H}$:

$$R(h) \leq \hat{R}_\gamma(h) + 2\mathfrak{R}((\mathcal{F}_\gamma)_{|S}) + 3\sqrt{\frac{\ln(2/\delta)}{2m}}, \quad (1)$$

where $\hat{R}_\gamma(h)$ represents the empirical margin loss, i.e., $\hat{R}_\gamma(h) = m^{-1}\sum_{i=1}^m \mathbb{1}(h(\rho_i)_{y_i} \leq \gamma + \max_{j \neq y_i} h(\rho_i)_j)$.

In this section, we present an analytic bound for $\mathfrak{R}((\mathcal{F}_\gamma)_{|S})$ in terms of quantum channel components, thereby establishing a margin-based generalization bound for QML. We focus on pure state inputs, $x \in \mathbb{C}^N$, which can be generalized to mixed state inputs via vectorization.

Given a set of POVMs $\{E_i\}_{i=1}^k$, we define the quantum measurement functions as $g_i(x) = x^\dagger E_i x$ and $g(x) = \{x^\dagger E_i x\}_{i=1}^k$. For pure state inputs, the function class becomes $\mathcal{F}_\gamma = \{(x, y) \mapsto l_\gamma(\mathcal{M}(g(Ux), y)) : U \in \mathbb{U}_Q\}$.

To derive an analytic expression for the Rademacher complexity $\mathfrak{R}((\mathcal{F}_\gamma)_{|S})$ in terms of the quantum channel components, we employ *covering number* techniques, which discretize the function class with a finite set of representative elements. The $\epsilon$-covering number of a set $A$, denoted as $\mathcal{N}(A, \epsilon, \|\cdot\|)$, represents the minimum cardinality of any subset $B \subseteq A$ such that $\sup_{a \in A} \min_{b \in B} \|a - b\| \leq \epsilon$. Obtaining a covering number bound for $(\mathcal{F}_\gamma)_{|S}$ involves two steps: (1) utilizing the Lipschitz continuity of $l_\gamma$, $\mathcal{M}$, and $g$, and (2) employing matrix covering techniques in the context of QNNs.

Let $\|\cdot\|_\sigma$ denote the spectral norm, and let $\|\cdot\|_{p,q}$ denote element-wise $(p, q)$ matrix norm. For any $y$ and $p \geq 1$, $l_\gamma(M(\cdot, y))$ is $2/\gamma$-Lipschitz in the $l_p$ norm (see Lemma A.3 of Bartlett et al. (2017)). Moreover, $g_i$ is $2\|E_i\|_\sigma$-Lipschitz, as $\|\nabla g_i(z)\|_2 \leq 2\|E_i\|_\sigma$ for all normalized quantum states $z$. Consequently,

$$\|g(u) - g(v)\|_2 \leq 2\sqrt{\sum_i \|E_i\|_\sigma^2}\|u - v\|_2,$$

showing that $g$ is $2E$-Lipschitz, where $E = \sqrt{\sum_i \|E_i\|_\sigma^2}$.

Lipschitz continuity reduces the covering numbers of $(\mathcal{F}_\gamma)_{|S}$ to matrix covering. For a data matrix $X \in \mathbb{C}^{N \times m}$, where each column represents a quantum state, the Frobenius norm satisfies $\|X\|_2 = \sqrt{m}$ due to normalization. Given a reference unitary $U_{\mathrm{ref}}$ and bound $b$ such that $\|U - U_{\mathrm{ref}}\|_{2,1} \leq b$ for all $U \in \mathbb{U}_Q$, we obtain

$$\mathcal{N}\left((\mathcal{F}_\gamma)_{|S}, \epsilon, \|\cdot\|_2\right) \leq \mathcal{N}\left(\{UX : U \in \mathbb{U}_Q\}, \frac{\epsilon\gamma}{4E}, \|\cdot\|_2\right)$$
$$\leq \exp\left(\left\lceil\frac{32mb^2E^2}{\epsilon^2\gamma^2}\right\rceil \ln 4N^2\right).$$

The first inequality follows from Lipschitz scaling, while the second adapts matrix covering techniques to our quantum setting, extending Bartlett et al. (2017) and Zhang (2002).

Finally, we bound the Rademacher complexity in terms of the covering number using the Dudley's entropy integral (Mohri et al., 2018; Shalev-Shwartz & Ben-David, 2014). Consequently, we have following margin bound for QNNs.

**Theorem 3.1.** *Consider an $n$-qubit QNNs consist of unitary $U \in \mathbb{U}_Q$ and POVMs $\{E_i\}_{i=1}^k$ for $k$-class classification. Let*

$b$ be a distance bound such that $\|U - U_{\text{ref}}\|_{2,1} \leq b$ holds for any $U \in \mathbb{U}_Q$, with $U_{\text{ref}}$ serving as the reference unitary matrix. Then, for any $\delta > 0$ and $\gamma > 0$, with probability at least $1 - \delta$ over the random draw of an i.i.d. sample $S$ of size $m$, the following inequality holds for all $h \in \mathcal{H}$:

$$R(h) \leq \hat{R}_\gamma(h) + \tilde{O}\left( \frac{b}{\gamma} \sqrt{\frac{n}{m} \sum_{i=1}^{k} \|E_i\|_\sigma^2} + \sqrt{\frac{\ln(1/\delta)}{m}} \right).$$

Unlike uniform generalization bounds, a margin bound offers a tighter upper bound when the hypothesis classifies the sample set $S$ with larger margins. Thus, the margin distribution, normalized by the spectral norms of POVMs, is critical for assessing generalization performance. For example, a left-skewed margin distribution, where many samples are classified with small margins, results in a large upper bound, indicating poor generalization performance.

Furthermore, when the POVMs are projective measurements (as in many QML models), the Lipschitz constant for quantum measurements simplifies to $2\sqrt{k}$. In this case, the margin bound simplifies as follows:

**Corollary 3.2.** *Under the conditions of Theorem 3.1, if the POVMs $\{E_i\}_{i=1}^{k}$ are projective measurements, where $E_i^2 = E_i$ and $E_i E_j = 0$ for all $i \neq j$, then:*

$$R(h) \leq \hat{R}_\gamma(h) + \tilde{O}\left( \frac{b}{\gamma} \sqrt{\frac{nk}{m}} + \sqrt{\frac{\ln(1/\delta)}{m}} \right).$$

This shows that, for QNNs with projective measurements, the margin bound is independent of the choice of measurement operators. Consequently, when comparing generalization performance across models with projective measurements, it suffices to evaluate the margin distribution alone, as normalization by the spectral norm is unnecessary due to its uniformity across models.

Details on the derivation of margin bound for QNNs and its extension to mixed input quantum states are provided in Appendix A.1.

## 4. Experimental Results

### 4.1. Margin Distribution and Generalization Performance

In this section, we empirically demonstrate a strong correlation between margin distribution and the generalization performance of QML models. Building on the framework presented by Gil-Fuster et al. (2024), we revisit the Quantum Phase Recognition (QPR) with randomized labels, utilizing Quantum Convolutional Neural Networks (QCNN) (Cong et al., 2019; Hur et al., 2022). QPR is a classification task aimed at identifying quantum phases of matter, a problem of significant relevance in condensed matter physics (Sachdev, 1999; 2023; Broecker et al., 2017; Ebadi et al., 2021; Carrasquilla & Melko, 2017).

Specifically, we examine the generalized cluster Hamiltonian defined as $H(J_1, J_2) = \sum_{j=1}^{n}(Z_j - J_1 X_j X_{j+1} - J_2 X_{j-1} Z_j X_{j+1})$, where $X_j$ and $Z_j$ are the Pauli operators acting on site $j$. This Hamiltonian has tunable parameters $J_1$ and $J_2$ that control interaction strengths. Depending on the values of these parameters, the ground state of the Hamiltonian falls into one of four distinct phases: (1) ferromagnetic, (2) antiferromagnetic, (3) symmetry-protected topological (SPT), or (4) trivial. Consequently, determining the phase of a given ground state with unknown interaction parameters is framed as a four-class classification problem.

QCNNs can overfit the randomized QPR labels, revealing the limitations of uniform bounds in assessing generalization performance (Gil-Fuster et al., 2024). We argue that the margin bound offers a more accurate measure of generalization in QML models, evidenced by the strong correlation between margin distribution and test accuracy.

Figure 1 presents the margin distributions of optimized 8-qubit QCNNs using box-and-whisker plots, along with corresponding test accuracies and generalization gaps, for models with one, five, and nine QCNN layers. A right-skewed box plot suggests that the data are classified with larger margins, which is associated with a tighter generalization upper bound (i.e., a smaller right-hand side in the equation of Theorem 3.1).

Across all layer configurations, increasing label randomization leads to decreased test accuracy. Concurrently, label randomization shifts the margin distributions significantly to the left. This leftward shift in the margin distributions results in a larger upper bound on the true error, as indicated in Theorem 3.1, showing that the margin bound effectively captures the generalization behavior under randomized labels.

Additionally, when labels are not completely randomized, QCNNs with deeper layers tend to achieve higher test accuracy. This suggests that deeper QCNNs, due to increased expressibility, can identify hypotheses closer to the optimal one. As the test accuracy increases, we observe a corresponding rightward shift in the margin distributions, further highlighting that margin distributions are reliable indicators of generalization performance in the QML framework.

Detailed experimental procedures and additional results using different variational ansätze, including more fine-grained analyses of noise levels and layer counts, are provided in Appendix A.3, Figures 20 and 21.

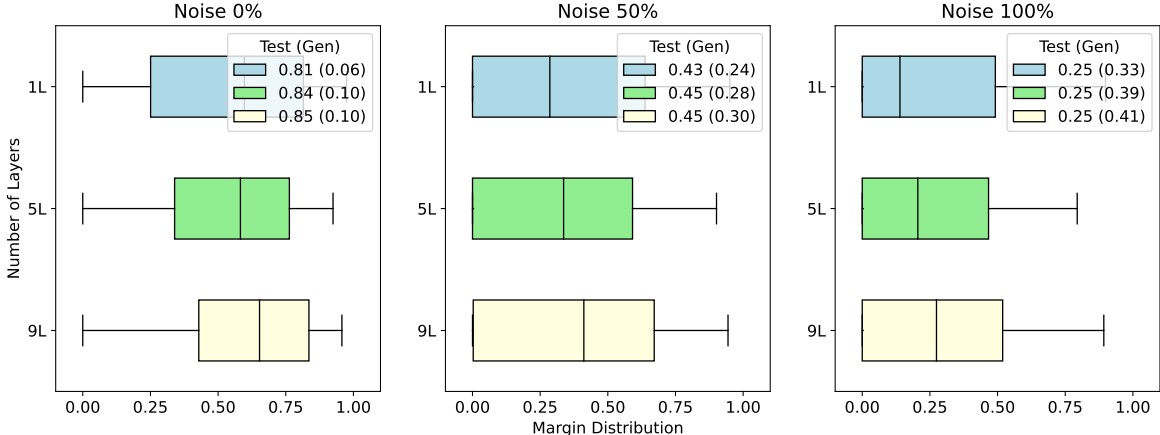

*Figure 1.* A Tukey box-and-whisker plot depicting the margin distributions of optimized 8-qubit Quantum Convolutional Neural Networks (QCNNs). The results for QCNNs with one, five, and nine layers are displayed, along with their corresponding test accuracies and generalization gaps indicated in the legend. QCNNs were trained for 4-class classification task aimed at quantum phase recognition (QPR). The experiment was performed with varying degrees of label noise: QPR dataset with pure labels (left), half randomly labelled dataset (middle), and full randomly labelled datasets (right). As the noise (randomization) level increases, the margin distributions tend to exhibit a more pronounced skew towards the left, indicating that a greater proportion of samples are classified with smaller margins. Notably, the margin distribution exhibits a strong positive correlation with test accuracy across all scenarios.

## 4.2. Predicting Generalization Gap: Parameters vs Margins

In our earlier analyses, we identified a strong correlation between margin distributions and generalization performance. Here, we demonstrate the effectiveness of margin-based metrics in estimating the generalization gap, highlighting their advantages over traditional uniform bounds.

Caro et al. (2022) showed that the generalization gap in QML models can be estimated based on the number of trainable parameters. Specifically, they proved that, in the worst case, the generalization gap scales with the square root of the parameter count. Furthermore, when only a subset of parameters undergoes substantial change during training, the generalization gap scales with the square root of the number of effective parameters that undergo significant updates. Despite being uniform bounds, the parameter-based generalization approach has become a standard method for understanding generalization behavior in QML models.

To illustrate the effectiveness of margins in estimating the generalization gap, we compare three margin-based metrics with three parameter-based metrics. For margin-based metrics, we analyze the lower quartile, median, and mean values of the margin distribution. For parameter-based metrics, we examine both the total number of parameters and the count of effective parameters that undergo substantial change during optimization. Specifically, we define effective parameters using two thresholds, $10^{-1}$ and $10^{-2}$, which represent the minimum change in a parameter (the rotation angle of parameterized quantum gates) for it to be considered effective.

Figure 2 depicts how the generalization gap, median margin, and effective parameters (with a $10^{-2}$ threshold) change in response to variations in the number of QCNN layers, the percentage of randomized labels, and the choice of variational ansatz. Since margins are inversely correlated with generalization gap (as shown in Theorem 3.1), the inverse of the median margin is plotted instead. Additionally, following the results of Caro et al. (2022), we plot the square root of the number of effective parameters.

Figure 2(a) shows how the generalization gap, inverse median margin, and the square root of effective parameters vary with the number of QCNN layers—1, 3, 5, 7, and 9. The generalization gap reaches its peak at five layers before slightly decreasing. While the effective parameters increase monotonically with the number of layers, the median margin effectively captures the peak at five layers.

Similarly, in Figure 2(b), the margin effectively captures the rising generalization gap as the percentage of randomized labels increases, while the number of effective parameters show an inverse trend.

In Figure 2(c), we analyze three variational ansätze: QCNN, QCNN with shared parameters (Cong et al., 2019; Hur et al., 2022), and Strongly Entangling Layers (Bergholm et al., 2020), arranged in decreasing order of expressibility. In this case, the QCNN with shared parameters constrains the local parameterized unitaries within the convolutional layers to share identical parameter values. The margin correctly reflects the rising generalization gap as expressibility de-

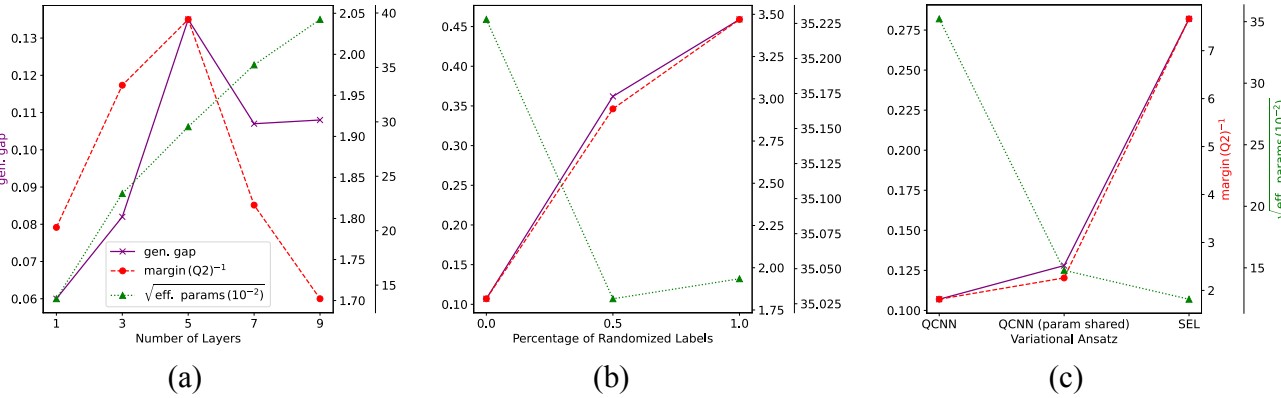

(a)  (b)  (c)

*Figure 2.* Illustration of how the generalization gap, median of the margin distribution (a margin-based metric), and effective parameters with a $10^{-2}$ threshold (a parameter-based metric) vary with the number of layers, percentage of randomized labels, and the choice of variational ansätze.

creases, while the number of effective parameters shows an inverse trend.

In summary, the margin reliably captures variations in the generalization gap across different hyperparameters, whereas effective parameters fail to do so and sometimes show the opposite trend. Note that the median margin is used instead of the lower quartile (Q1) to avoid infinite inverse margins when Q1 is zero, a scenario that occasionally arises when the labels are fully randomized and the model employs an inexpensive variational ansatz. Effective parameters are shown instead of the total number of parameters, as the latter remains constant despite changes in randomized label percentages, unlike the generalization gap. Additional comparisons—margin mean and Q1 versus total and effective parameters—without label noise are provided in Appendix A.3, Figure 22.

Thus far, we have compared how margins and the number of parameters capture generalization by varying one hyperparameter at a time while keeping the others fixed. Now, we present a more comprehensive comparison with all hyperparameters varied simultaneously. More specifically, for a given set of hyperparameters $\lambda$, we examine the correlation between the generalization gap $g(\lambda)$ and the corresponding metric $\mu(\lambda)$. Here, as before, the metrics can be either margin-based or parameter-based. To assess correlation, we analyzed two different evaluation methods: mutual information and Kendall rank correlation coefficient. Both methods (or their variants) have been used to assess generalization capacity in classical deep learning (Jiang et al.; 2020; Dziugaite et al., 2020).

In mutual information analysis, we treat the hyperparameter $\lambda$ as a random vector, with $g(\lambda)$ and $\mu(\lambda)$ as functions of this vector. Mutual information, denoted by $I(g; \mu)$, is defined as $I(g; \mu) = \mathbb{E}\left[\log \frac{P(g,\mu)}{P(g)P(\mu)}\right]$, which simplifies to

$H(g) - H(g|\mu)$. Here, $H(g)$ represents the entropy of $g$, quantifying the uncertainty of the generalization gap, while $H(g|\mu)$ is the conditional entropy of $g$ given $\mu$, indicating the remaining uncertainty about the generalization gap given $\mu$. Thus, higher mutual information $I(g; \mu)$ suggests that $\mu$ provides more information about $g$, reducing the uncertainty associated with $g$.

In contrast, the Kendall rank correlation coefficient $\tau$ measures the strength and direction of the association between two variables. For pairs of generalization gap and metric values, $(g(\lambda_1), \mu(\lambda_1))$ and $(g(\lambda_2), \mu(\lambda_2))$, we expect the metric to accurately capture the ranking of generalization performance. Specifically, if $g(\lambda_1) < g(\lambda_2)$, then $\mu(\lambda_1)$ should also be less than $\mu(\lambda_2)$. This implies that the metric $\mu$ effectively predicts relative generalization, with lower $\mu$ values corresponding to better generalization (lower $g$). Consider lists of $n$ generalization gaps and corresponding metrics, denoted as $G = [g_1, \ldots, g_n]$ and $M = [\mu_1, \ldots, \mu_n]$, where each $g_i$ and $\mu_i$ corresponds to $g(\lambda_i)$ and $\mu(\lambda_i)$, respectively. The Kendall rank correlation coefficient between $G$ and $M$ is given by

$$\tau_{G,M} = \frac{1}{2n(n-1)} \sum_{i<j} \left[1 + \text{sgn}(g_i - g_j)\,\text{sgn}(\mu_i - \mu_j)\right].$$

This coefficient ranges from 0 to 1, where $\tau = 1$ indicates perfect agreement between the rankings of $g$ and $\mu$ (all pairs are concordant), and $\tau = 0$ indicates perfect disagreement (all pairs are discordant).

Figure 3 shows the mutual information (solid) and Kendall rank correlation coefficient (shaded) between the generalization gap and various metrics. In both evaluation methods, margin-based metrics demonstrate stronger correlations than parameter-based metrics, indicating that margins are more effective in evaluating the generalization performance of QML models.

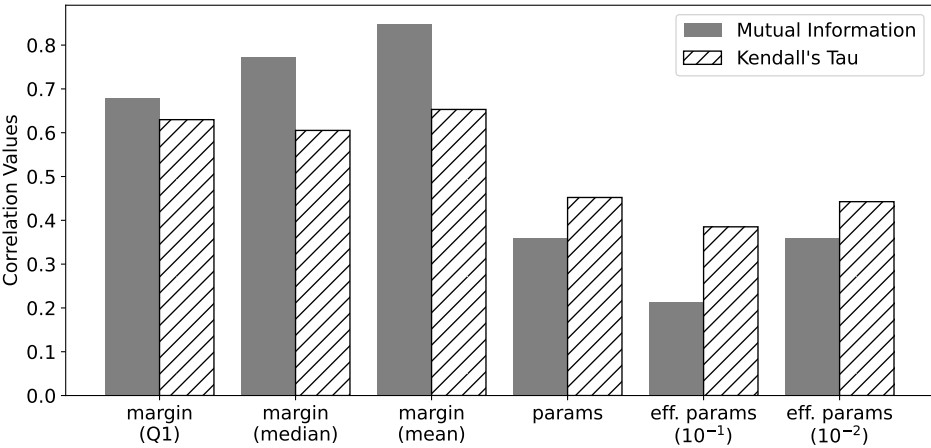

*Figure 3.* Comparative analysis of mutual information (solid) and Kendall rank correlation coefficients (shaded) between the generalization gap and various metrics. The first three columns represent margin-based metrics, while the last three columns represent parameter-based metrics.

Consistent with previous analyses, we used inverse margin values to measure correlation, as larger margins correspond to smaller generalization gaps. Further details on this experiment can be found in Appendix A.2.

To further demonstrate the robustness of our results, we present additional experimental results that reproduce the main findings (Figures 1-3). These experiments were conducted on supplementary datasets—the Transverse Field Ising Model and the XXZ Heisenberg spin chain—with 8 and 10 qubits. These results are detailed in Appendix A.3.

## 5. Margins and Quantum Embedding

Up to this point, we have focused on scenarios involving the classification of $n$-qubit quantum data using QML models. Another important application of QML is the classification of classical data. To work with classical data in a quantum framework, the $d$-dimensional data, $x \in \mathbb{R}^d$, must be mapped to an $n$-qubit quantum state, represented by a density matrix $\rho(x) \in \mathbb{C}^{2^n \times 2^n}$. This process, known as *quantum embedding*, typically involves applying a quantum embedding circuit $U_{\text{emb}}$ to the ground state, resulting in $\rho(x) = U_{\text{emb}}(x)(|0\rangle\langle 0|)^{\otimes n} U_{\text{emb}}^\dagger(x)$.

Quantum embedding is crucial to the overall performance of QML models, as it establishes the lower bound of the training loss (Lloyd et al., 2020; Hur et al., 2024). For example, in a binary classification with a dataset of $N$ samples, $S = \{x_i^-, -1\}_{i=1}^{N^-} \cup \{x_i^+, +1\}_{i=1}^{N^+}$, the training loss with respect to a linear loss function, denoted by $L_S$, is lower bounded as

$$L_S \geq \frac{1}{2} - D_{\text{tr}}(p^- \rho^-, p^+ \rho^+).$$

Here, $\rho^\pm = \sum_i \rho(x_i^\pm)/N^\pm$ represents the quantum states corresponding to the ensemble of each class. The probabilities for each class are denoted by $p^\pm = N^\pm/N$, and $D_{\text{tr}}(\cdot, \cdot)$ denotes the trace distance.

Given a sample dataset $S$, the quantum embedding circuit determines $D_{\text{tr}}(p^- \rho^-, p^+ \rho^+)$. A quantum embedding that produces a large initial trace distance results in a smaller loss, and vice versa. This is because quantum channels are completely positive and trace-preserving (CPTP) maps, which cannot increase the distinguishability between quantum states. Formally, for a quantum channel $\Lambda$ and a distance metric $D$ (such as trace distance or infidelity), the contractive property $D(\Lambda(\rho^+), \Lambda(\rho^-)) \leq D(\rho^+, \rho^-)$ applies to any quantum states $\rho^+$ and $\rho^-$. This indicates that the distance between two quantum states (representing data from different classes) cannot increase as quantum operations are applied, emphasizing the importance of starting with a large initial trace distance.

To address this limitation, Lloyd et al. (2020) introduced trainable quantum embedding (TQE), where a parameterized quantum circuit is optimized to maximize the separation between data classes in Hilbert space. Unlike a fixed quantum embedding, the $L$-layer TQE incorporates an additional parameterized unitary circuits, $V(\phi)$, to construct $U_{\text{tqe}}(x, \phi) = \prod_{i=1}^L U_{\text{emb}}(x)V(\phi_i)$. The quantum state associated with data $x$ and parameter $\phi$ is given by $\rho(x, \phi) = U_{\text{tqe}}(x, \phi)(|0\rangle\langle 0|)^{\otimes n} U_{\text{tqe}}^\dagger(x, \phi)$. The trainable parameters $\phi$ are optimized to enhance the distinguishability between quantum states corresponding to different data classes. This concept was later extended to quantum kernel frameworks by Hubregtsen et al. (2022).

Building on this, Hur et al. (2024) proposed Neural Quan-

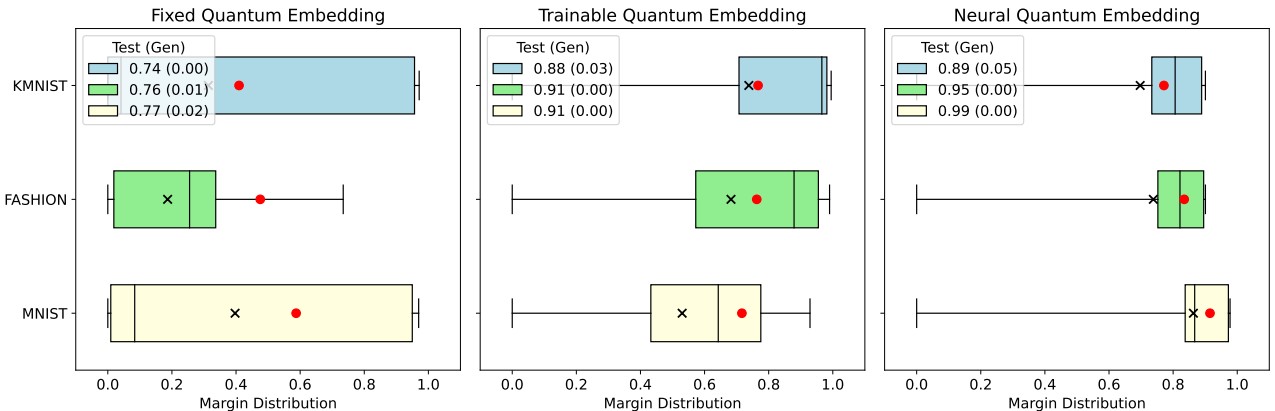

*Figure 4.* A Tukey box-and-whisker plot illustrating the margin distributions of optimized 8-qubit Quantum Convolutional Neural Networks (QCNNs). The plot shows results for QCNNs using fixed quantum embedding (left), trainable quantum embedding (middle), and neural quantum embedding (right). The QCNNs were trained on a binary classification task using the MNIST (bottom), Fashion-MNIST (middle), and Kuzushiji-MNIST (top) datasets. In addition to the margin distributions, the mean of the margins is indicated by a black cross, and the trace distance between ensemble quantum states is marked by a red circle.

tum Embedding (NQE), which leverages classical neural networks to optimize quantum embeddings. Their study highlighted the inherent limitations of TQE in maximizing trace distance and empirically demonstrated that NQE can effectively identify quantum embeddings with a large initial trace distance, outperforming the capabilities of TQE.

While both TQE and NQE are theoretically designed to improve the lower bound of sample training loss, they have also been shown to significantly enhance test accuracy, thereby improving generalization performance. Although a positive relationship between a large initial trace distance and improved generalization has been observed empirically, a theoretical explanation for this relationship was previously missing. Margin generalization offers this missing theoretical basis, as the trace distance serves as an upper bound on the margin mean.

In binary classification, the margin mean simplifies to $\bar{\mu} = \frac{2}{N} \sum_i \mathrm{Tr}(U\rho(x_i)U^\dagger E_{y_i}) - 1$, where $U$ denotes the optimized unitary of QNN after the training. By defining $E_{\pm 1}^* = U^\dagger E_{\pm 1} U$, the margin mean can be reformulated as,

$$
\begin{aligned}
\bar{\mu} &= \frac{2}{N} \left[ \sum_{i=1}^{N^+} \mathrm{Tr}(\rho(x_i^+)E_{+1}^*) + \sum_{i=1}^{N^-} \mathrm{Tr}(\rho(x_i^-)E_{-1}^*) \right] - 1 \\
&= 2\mathrm{Tr}(p^+ \rho^+ E_{+1}^*) + 2\mathrm{Tr}(p^- \rho^- E_{-1}^*) - 1 \\
&\leq D_{\mathrm{tr}}(p^+ \rho^+, p^- \rho^-), \quad\quad\quad (2)
\end{aligned}
$$

where the final inequality becomes equality if and only if $\{E_{\pm 1}^*\}$ corresponds to the Helstrom measurement (see Appendix A.4). In other words, TQE and NQE enhance the model's generalization capability by identifying quantum

embeddings with a large trace distance, thereby increasing the upper bound of the margin mean. This margin-based perspective on generalization enables a systematic optimization of the generalization performance of QML models by selecting quantum embeddings with large trace distances.

Figure 4 presents the classification of MNIST (LeCun et al., 2010), Fashion-MNIST (Xiao et al., 2017), and Kuzushiji-MNIST (Clanuwat et al., 2018) datasets using 8-qubit QCNN with various quantum embedding schemes. The comparison highlights how the initial trace distance affects the margin distribution and the model's generalization performance. Three quantum embedding schemes are employed to vary the initial trace distance: the fixed quantum embedding (ZZ Feature Map introduced by Havlícek et al. (2019)), TQE, and NQE.

As anticipated from previous research, we observed a progressively increasing trace distance (indicated by a red dot) in the order of fixed quantum embedding, TQE, and NQE. This increase in trace distance corresponds to higher margin mean values (indicated by a black cross). Larger margin means, coupled with right-skewed margin distributions, are associated with improved test accuracies, aligning with the principles of margin-based generalization. In summary, using an optimal quantum embedding with a large initial trace distance not only reduces sample training loss but also results in larger margins and enhanced generalization. Experimental details are provided in Appendix A.2.

## 6. Discussion

In this work, we proposed a margin-based framework to better understand generalization in QML models. We began by

addressing the limitations of uniform generalization bounds, which have proven ineffective when models overfit noisy labels in both classical and quantum settings. Drawing inspiration from margin-based approaches in deep learning, we developed a margin generalization bound for QNNs, offering tighter and more meaningful estimates of generalization performance.

Through extensive empirical studies, we demonstrated that margin-based metrics are highly predictive of generalization performance in QML models. Our experiments with QCNNs on the QPR dataset revealed a strong correlation between margin distributions and generalization. Furthermore, we compared margin-based metrics against parameter-based metrics for predicting the generalization gap. In all settings, margin-based metrics proved to be better predictors of the generalization gap, showing higher mutual information and Kendall rank correlation coefficients.

Additionally, we explored the connection between margin-based generalization and quantum state discrimination, a fundamental concept in quantum information theory. We showed that quantum embeddings with a large trace distance lead to larger margins, which in turn improve generalization. By comparing different quantum embedding methods, we experimentally demonstrated that embeddings with higher trace distances consistently result in better generalization performance.

Overall, our results suggest that margin-based metrics provide a more reliable and interpretable framework for understanding generalization in QML models. This framework not only offers theoretical insights but also practical guidance for designing QML architectures that generalize well.

We highlight several open questions for future research. First, quantum computers without error correction are highly susceptible to noise and decoherence. Quantum noise, modeled by CPTP maps, reduces trace distance between quantum states, leading to smaller classification margins and poorer generalization performance. A deeper theoretical understanding of the relationship between noise, margins, and generalization, along with strategies to mitigate these effects, will be critical for the effective use of QML in noisy intermediate-scale quantum devices.

Furthermore, exploring generalization from the perspective of the *learning algorithm* would offer valuable insights (Berberich et al., 2024). The algorithm $\mathcal{A}$, whether classical or quantum, processes a training set $S$ to produce a hypothesis $h$. While traditional uniform bounds focus on the worst-case scenario across the entire hypothesis class, it is more practical to examine the generalization gap of the specific hypothesis learned by the algorithm, $R(\mathcal{A}(S)) - \hat{R}(\mathcal{A}(S))$. In classical machine learning, research has explored this through *algorithmic stabil-*

*ity*, assessing how sensitive $\mathcal{A}$ is to changes in the training data (Bousquet & Elisseeff, 2002; Neyshabur et al., 2017; Dziugaite & Roy, 2017). Another promising approach explores the relationship between generalization and *optimization speed*, such as the number of iterations required to reach a certain training loss (Hardt et al., 2016). Extending these lines of research to QML could provide deeper insights into the generalization behavior of quantum models.

## Acknowledgments

This work was supported by Institute for Information & communications Technology Promotion (IITP) grant funded by the Korea government (No. 2019-0-00003, Research and Development of Core technologies for Programming, Running, Implementing and Validating of Fault-Tolerant Quantum Computing System), the Yonsei University Research Fund of 2024 (2024-22-0147), the National Research Foundation of Korea (2023M3K5A109481312), and the Ministry of Trade, Industry, and Energy (MOTIE), Korea, under the Industrial Innovation Infrastructure Development Project (Project No. RS-2024-00466693).

## Impact Statement

This paper presents work whose goal is to advance the field of Quantum Machine Learning. There are many potential societal consequences of our work, none of which we feel must be specifically highlighted here.

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

# A. Appendix

## A.1. Derivation of Margin Bound and Generalization to Mixed States

In Section 3, we present a theoretical generalization bound for Quantum Neural Networks (QNNs) using margin-based approaches. This bound quantifies how well a QNN model trained on a given dataset will perform on new, unseen samples. In this appendix, we expand on the derivation of this bound, offering additional insights into the steps and calculations involved.

To derive the upper bound, we apply Rademacher complexity to the hypothesis classes of QNNs that incorporate margin loss. We proceed by utilizing covering number arguments, which enable further refinement of the upper bound on sample Rademacher complexity. Specifically, we employ Dudley's Entropy Integral, a technique that converts the bound on Rademacher complexity into a function of covering numbers.

Our derivation relies on three established lemmas, summarized below without proofs. For further details, proofs of Lemma A.1 and Lemma A.3 can be found in Shalev-Shwartz & Ben-David (2014) and Mohri et al. (2018), while proof of Lemma A.2 can be found in Zhang (2002) and Bartlett et al. (2017).

**Lemma A.1.** *(Rademacher Complexity) Let $\mathcal{F}$ be a class of functions where each $f(z)$ takes values within an interval $[0, 1]$. Then, for any $\delta > 0$, with probability at least $1 - \delta$ over the random draw of an i.i.d. sample $S$ of size $m$, the following inequality holds for all $f \in \mathcal{F}$:*

$$\sup_{f \in \mathcal{F}} \mathbb{E}\left[f(Z)\right] \leq \frac{1}{m} \sum_{i=1}^{m} f(z_i) + 2\mathfrak{R}(\mathcal{F}_{|S}) + 3\sqrt{\frac{\ln(2/\delta)}{2m}}. \tag{3}$$

Lemma A.1 provides an upper bound on the expectation of any function within $\mathcal{F}$ using sample Rademacher complexity $\mathfrak{R}(\mathcal{F}_{|S})$.

**Lemma A.2.** *(Maurey's Sparsification Lemma) Consider a Hilbert space $\mathcal{H}$ with a norm $\|\cdot\|$ and a set of vectors $V = \{V_1, \ldots, V_d\}$ where each $V_i \in \mathcal{H}$. Define a scaled convex combination $U = \alpha \cdot \mathrm{conv}(V)$, where $\alpha$ is a positive real constant, and $\mathrm{conv}(V)$ denotes the convex hull of $V$. For any integer $k$, there exists a vector $(k_1, \ldots, k_d) \in \mathbb{Z}_+^d$ such that $\sum_{i=1}^{d} k_i = k$, and*

$$\left\| U - \frac{\alpha}{k} \sum_{i=1}^{d} k_i V_i \right\|^2 \leq \frac{\alpha^2}{k} \max_i \|V_i\|^2. \tag{4}$$

Lemma A.2 provides an error bound for approximating a vector using a linear combination of discrete coefficients.

**Lemma A.3.** *(Dudley's Entropy Integral) Given a set $U \subseteq [-1, +1]^m$ that contains the origin, the Rademacher complexity $\mathfrak{R}(U)$ can be bounded by:*

$$\mathfrak{R}(U) \leq \inf_{\alpha > 0} \left( \frac{4\alpha}{\sqrt{m}} + \frac{12}{m} \int_{\alpha}^{\sqrt{m}} \sqrt{\ln \mathcal{N}(U, \beta, \|\cdot\|_2)} d\beta \right). \tag{5}$$

Lemma A.3 leverages covering numbers $\mathcal{N}(U, \beta, \|\cdot\|_2)$ to bound the Rademacher complexity through an integral over scales $\beta$, with $\alpha$ acting as an adjustable parameter to tighten the bound.

Applying Lemma A.1 to the function class $\mathcal{F}_\gamma = \{(\rho, y) \mapsto l_\gamma(\mathcal{M}(h(\rho), y)) : h \in \mathcal{H}\}$, we establish a bound on the expected margin loss in terms of the empirical sample. With probability at least $1 - \delta$, for all $h \in \mathcal{H}$, we have:

$$\mathbb{E}\left[l_\gamma(\mathcal{M}(h(\rho), y))\right] \leq \frac{1}{m} \sum_{i=1}^{m} l_\gamma(\mathcal{M}(h(\rho_i), y_i)) + 2\mathfrak{R}((\mathcal{F}_\gamma)_{|S}) + 3\sqrt{\frac{\ln(2/\delta)}{2m}}. \tag{6}$$

Since the ramp loss is lower bounded by the indicator function, $\mathbb{1}(x \leq 0) \leq l_\gamma(x)$, the expected loss is directly related to the misclassification probability: $\mathbb{E}\left[\mathbb{1}(h(\rho)_y \leq \max_{j \neq y} h(\rho)_j)\right] \leq \mathbb{E}[l_\gamma(\mathcal{M}(h(\rho), y))]$. Similarly, because $l_\gamma(x) \leq \mathbb{1}(x \leq \gamma)$, the sample loss is bounded above by the empirical margin loss: $m^{-1} \sum_{i=1}^{m} l_\gamma(\mathcal{M}(h(\rho_i), y_i)) \leq m^{-1} \sum_{i=1}^{m} \mathbb{1}(h(\rho_i)_{y_i} \leq \gamma + \max_{j \neq y_i} h(\rho_i)_j)$. Substituting these relationships into the bound from Equation 6 yields Equation 1.

Equation 1 includes the Rademacher complexity term $\mathfrak{R}((\mathcal{F}_\gamma)_{|S})$, which we analyze using covering number techniques. Specifically, we are interested in the $\epsilon$-covering number of $(\mathcal{F}_\gamma)_{|S}$ with respect to the $l_2$ norm, denoted by $\mathcal{N}((\mathcal{F}_\gamma)_{|S}, \epsilon, \|\cdot\|_2)$.

For pure input states, the function class can be expressed as $\mathcal{F}_\gamma = \{(x, y) \mapsto l_\gamma(\mathcal{M}(g(Ux), y)) : U \in \mathbb{U}_Q\}$, where $g(x) = \{x^\dagger E_i x\}_{i=1}^k$ and $x \in \mathbb{C}^N$. As shown in the main text, $l_\gamma(\mathcal{M}(g(\cdot), y))$ is $4E/\gamma$-Lipschitz with respect to the $l_2$ norm, where $E = \sqrt{\sum_i \|E_i\|_\sigma}$.

For any $L$-Lipschitz function $f$ over a set $S$, we have the following covering number bound: $\mathcal{N}(f(S), \epsilon, \|\cdot\|) \le \mathcal{N}(S, \epsilon/L, \|\cdot\|)$. Applying this property, we obtain

$$\ln \mathcal{N}\left((\mathcal{F}_\gamma)_{|S}, \epsilon, \|\cdot\|_2\right) \le \ln \mathcal{N}\left(\{UX : U \in \mathbb{U}_Q\}, \frac{\gamma\epsilon}{4E}, \|\cdot\|_2\right), \tag{7}$$

where $X$ is $N \times m$ complex valued matrix whose $i$-th column represents the quantum state $x_i$.

The next step is to bound the covering number for the set $\{UX : U \in \mathbb{U}_Q\}$. Since the Frobenius norm is invariant under conjugate transposition, we can write $\mathcal{N}(\{UX : U \in \mathbb{U}_Q\}, \epsilon, \|\cdot\|_2) = \mathcal{N}(\{X^\dagger U^\dagger : U \in \mathbb{U}_Q\}, \epsilon, \|\cdot\|_2)$.

Define $Y \in \mathbb{C}^{m \times N}$ by normalizing each column of $X^\dagger$ to have unit $l_2$-norm: $Y_{:,j} = X_{:,j}^\dagger / \|X_{:,j}^\dagger\|_2$. Also, let $\alpha \in \mathbb{C}^{N \times N}$ be a scaling matrix where each element in the $i$-th row is set to $\|X_{:,i}^\dagger\|_2$. Then, we can express $X^\dagger U^\dagger$ as $Y(\alpha \odot U) = YB$, where $\odot$ denotes the Hadamard product (element-wise multiplication) of matrices.

Consider the set $V = \{V_1, \ldots, V_{4N^2}\} = \{gYe_je_k^\mathrm{T} : g \in \{+1, -1, +i, -i\}, \ j \in [N], \ k \in [N]\}$. For any matrix $B$, we can express $YB$ as

$$YB = \sum_{j,k} B_{jk} Y e_j e_k^\mathrm{T} = \|B\|_* \cdot \mathrm{conv}(V),$$

where $\|B\|_*$ is defined as $\sum_{j,k} (|\mathrm{Re}(B_{jk})| + |\mathrm{Im}(B_{jk})|)$.

For a distance bound $b$ such that $\|U\|_{2,1} \le b$ for all $U \in \mathbb{U}_Q$ (which can be generalized to $\|U - U_{\mathrm{ref}}\|_{2,1} \le b$ for arbitrary reference matrix $U_{\mathrm{ref}}$), $\|B\|_*$ can be bounded by

$$\|B\|_* \le \sqrt{2} \sum_{j,k} |B_{jk}| \le \sqrt{2} \|U\|_{2,1} \|\alpha\|_{2,\infty} \le b\sqrt{2m},$$

where the first inequality follows from the vector norm inequality, the second from Holder's inequality, and the final inequality from the fact that $\|\alpha\|_{2,\infty} = \|X\|_2 = \sqrt{m}$.

For any $\epsilon > 0$, by setting $k = \lceil \frac{2mb^2}{\epsilon^2} \rceil$ and applying Lemma A.2, we conclude that there exists a vector $(k_1, \ldots, k_{4N^2}) \in \mathbb{Z}_+^d$ with $\sum_i k_i = k$ such that

$$\left\| X^\dagger U^\dagger - \frac{\|B\|_*}{k} \sum_i k_i V_i \right\|_2^2 \le \frac{\|B\|_*^2}{k} \max_i \|V_i\|^2 \le \epsilon^2. \tag{8}$$

Consequently, the set $C = \left\{ \frac{\|B\|_{1*}}{k} \sum k_i V_i : k \ge \lceil \frac{2mb^2}{\epsilon^2} \rceil \right\}$ serves as an $\epsilon$-cover for $\{X^\dagger U^\dagger : U \in \mathbb{U}_Q\}$ with respect to $l_2$-norm. Since $|C| \le 4N^{2\lceil \frac{2mb^2}{\epsilon^2} \rceil}$, it follows that

$$\ln \mathcal{N}\left(\{UX : U \in \mathbb{U}_Q\}, \epsilon, \|\cdot\|_2\right) \le \left\lceil \frac{2mb^2}{\epsilon^2} \right\rceil \ln 4N^2. \tag{9}$$

Combining this with Equation 7, we obtain

$$\ln \mathcal{N}((\mathcal{F}_\gamma)_{|S}, \epsilon, \|\cdot\|_2) \le \left\lceil \frac{32mb^2E^2}{\epsilon^2\gamma^2} \right\rceil \ln 4N^2. \tag{10}$$

Next, we bound the Rademacher complexity term using covering numbers via Dudley's Entropy Integral. Applying Lemma A.2 yields,

$$\mathfrak{R}((\mathcal{F}_\gamma)_{|S}) \leq \inf_{\alpha > 0} \left( \frac{4\alpha}{\sqrt{m}} + \frac{12}{m}\sqrt{2\ln 4N}\sqrt{\left\lceil \frac{32mb^2 E^2}{\gamma^2} \right\rceil}\ln\left(\frac{\sqrt{m}}{\alpha}\right) \right). \tag{11}$$

Setting $\alpha = \sqrt{\frac{18\ln 4N}{m}\left\lceil \frac{32mb^2 E^2}{\gamma^2} \right\rceil}$, we find

$$\mathfrak{R}((\mathcal{F}_\gamma)_{|S}) \leq \frac{12}{m}\sqrt{2\ln 4N}\sqrt{\left\lceil \frac{32mb^2 E^2}{\gamma^2} \right\rceil} \in \tilde{O}\left( \frac{bE}{\gamma}\sqrt{\frac{n}{m}} \right). \tag{12}$$

This yields the margin-based generalization bound (Theorem 3.1) in terms of the QNN components.

The margin bound for QNNs can be extended to mixed quantum states via vectorization. Given a mixed quantum state $\rho = \sum_{i,j}\rho_{ij}|i\rangle\langle j|$, we consider its vectorized form as a pure state: $|\rho\rangle\rangle = \sum_{i,j}\rho_{ij}|i\rangle \otimes |j\rangle$. In this framework, the measurement function $g_i$ (corresponding to the POVM element $E_i$) for the input mixed state is given by $g_i(\rho) = \text{Tr}(E_i\rho) = \langle\langle E_i|\rho\rangle\rangle$. Since $\|\nabla g_i(\rho)\|_2 = \|E_i\|_2$, the function $g_i$ is $\|E_i\|_2$-Lipschitz. Therefore, following the derivation in the main text, the combined measurement function $g$ is $E$-Lipschitz, where $E = \sqrt{\sum_i \|E_i\|_2^2}$.

For the mixed input state, the function class $\mathcal{F}_\gamma$ is defined as $\mathcal{F}_\gamma = \{(\rho, y) \mapsto l_\gamma(\mathcal{M}(g(U\rho U^\dagger)), y)\}$. Using the property $|U\rho U^\dagger\rangle\rangle = (U \otimes U^*)|\rho\rangle\rangle$, we extend the margin bound from Theorem 3.1 to mixed input states. The extension involves a slight adjustment: the distance bound $b$ now applies to $U \otimes U^*$ rather than $U$, and $E = \sqrt{\sum_i \|E_i\|_2^2}$ is evaluated using the Frobenius norm, unlike the spectral norm used for pure input states.

### A.2. Experimental Details

QUANTUM PHASE RECOGNITION: MARGIN DISTRIBUTION

In Section 4.1, 8-qubit quantum convolutional neural networks (QCNNs) were trained to address the quantum phase recognition (QPR) task using a generalized cluster Hamiltonian. The QCNNs employed nearest-neighbor two-qubit parameterized quantum circuits (PQCs). For this experiment, we used the architecture proposed in Hur et al. (2022), where the two-qubit PQCs follow the structure outlined in Figure 2 (i) of the reference.

The model was trained on 20 data points, evenly split across four classes. A small training sample was deliberately chosen to explore the overfitting regime, where labels were intentionally randomized with noise, following a methodology similar to that of Gil-Fuster et al. (2024). The test accuracy was measured on 1,000 test samples—far exceeding the size of the training set—to provide a robust estimate of true accuracy. The model was trained using Adam optimizer, with a learning rate of 0.001 and full-batch updates (Kinga et al., 2015). The model was trained for up to 5,000 iterations, with early stopping triggered based on a convergence interval of 500 iterations. Specifically, the training halted when the difference between the average loss over two consecutive intervals became smaller than the standard deviation of the most recent interval. To ensure reliable margin distribution boxplots given the small training set, the results were averaged over 15 experimental repetitions, each using a different training sample.

QUANTUM PHASE RECOGNITION: MARGIN VS PARAMTER

In Section 4.2, we compare parameter-based metrics against margin-based metrics for predicting the generalization gap. The experiment was conducted under the same conditions as described above, except for the convergence interval, which was set to 300 iterations.

For the mutual information calculation, the empirical joint probability distribution was used for the generalization gap $g$ and metrics $\mu$ to approximate the unknown true distribution. For Kendall rank correlation coefficient calculation, random tie-breakers were used when equal generalization gap or metrics occurred.

Figure 2 compares the median of the margin distribution with the effective number of parameters (threshold $10^{-2}$) in predicting the generalization gap. The experiments vary a single hyperparameter—either the number of layers, noise level, or variational ansatz—while keeping the others fixed at seven layers, no noise, and QCNN.

MARGINS AND QUANTUM EMBEDDING

In Section 5, we explore the influence of quantum embeddings and initial trace distances on margin and generalization performance when applying QML models to classical data. To maintain a binary classification setting, we focused on the first two classes from the MNIST, fashion-MNIST, and Kuzushiji-MNIST datasets. Unlike previous experiments, we did not examine the overfitting regime under label noise, choosing instead to utilize the full training and test datasets. Principal component analysis (PCA) was applied to reduce the dimensionality of the data to match the number of qubits. The experimental setup remained the same as before, except for using a batch size of 16 instead of full-batch training.

We compared three quantum embedding methods, each resulting in distinct initial trace distances. For the fixed embedding, we used the ZZ Feature Map with three repeated layers. The ZZ Feature Map consists of blocks of Hadamard gates, single-qubit Pauli-Z rotations, and two-qubit Pauli-ZZ rotations as described in Havlícek et al. (2019).

For the trainable quantum embedding, we introduced trainable quantum layers along with the ZZ Feature Map. The trainable quantum layers incorporate parameterized single-qubit Pauli-$Y$ rotations, $\exp(-i\frac{\theta}{2}Y_j)$, and nearest-neighbor two-qubit $YY$ rotations, $\exp(i\frac{\theta}{2}Y_jY_{j+1})$.

Neural Quantum Embedding (NQE) utilizes a classical neural network, $\sigma(x;w)$, where $x$ represents the data and $w$ the network parameters. NQE maps classical data into quantum states as $\rho(x) = U_{\text{emb}}(\sigma(x;w))|0\rangle\langle 0|^{\otimes n}U_{\text{emb}}^{\dagger}(\sigma(x;w))$. Here, the ZZ Feature Map was also employed as the quantum embedding circuit. The neural network was a fully-connected ReLU network with layer dimensions of $[8, 16, 32, 32, 16, 8]$, where each element represents the number of nodes in that layer.

### A.3. Additional Experimental Results

This section details supplementary experimental investigations, where we replicated the core experiments presented in Figures 1-3 of the main text. These replications involved larger 10-qubit systems and a more diverse set of Hamiltonians to further validate the robustness of our methodology. In addition to the generalized cluster Hamiltonian (detailed in the main text), our analysis includes the one-dimensional (1D) XXZ-Heisenberg spin chain and the 1D Transverse Field Ising Model (TFIM).

### XXZ-Heisenberg Spin Chain

The Hamiltonian for this model is defined as:

$$H(J, \Delta) = J\left(\sum_{i=1}^{N} X_iX_{i+1} + Y_iY_{i+1} + \Delta Z_iZ_{i+1}\right) \tag{13}$$

In our simulations, we set the coupling strength $J = 1$. The anisotropy parameter $\Delta$ was sampled uniformly from the combined intervals $[-2, -1) \cup (1, 2]$. This configuration leads to a ground state that is either ferromagnetic (when $\Delta < -1$) or antiferromagnetic (when $\Delta > 1$). QML models were subsequently trained to perform binary classification between these two distinct phases.

### Transverse Field Ising Model (TFIM)

The TFIM Hamiltonian is given by:

$$H(J, \Delta) = -J\left(\sum_{i=1}^{N} Z_iZ_{i+1} + \Delta \sum_{i=1}^{N} X_i\right) \tag{14}$$

For dataset construction, both the coupling $J$ and the transverse field strength $\Delta$ were sampled from uniform distributions. Our investigation focused on the ordered phase, characterized by $|\Delta| < 1$. The coupling $J$ was varied within the range $[-1, 1]$, allowing the system to exhibit either a ferromagnetic ($J > 0$) or an antiferromagnetic ($J < 0$) phase. A QML model was then trained for the binary classification task of distinguishing these phases.

10-QUBIT GENERALIZED CLUSTER HAMILTONIAN

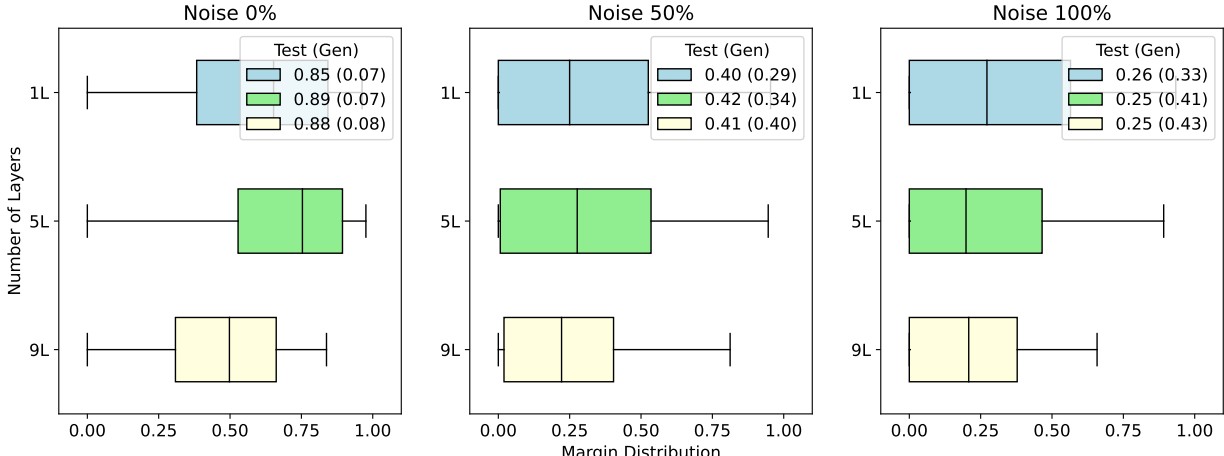

*Figure 5.* Reproduction of Figure 1 with 10-qubit generalized cluster Hamiltonian.

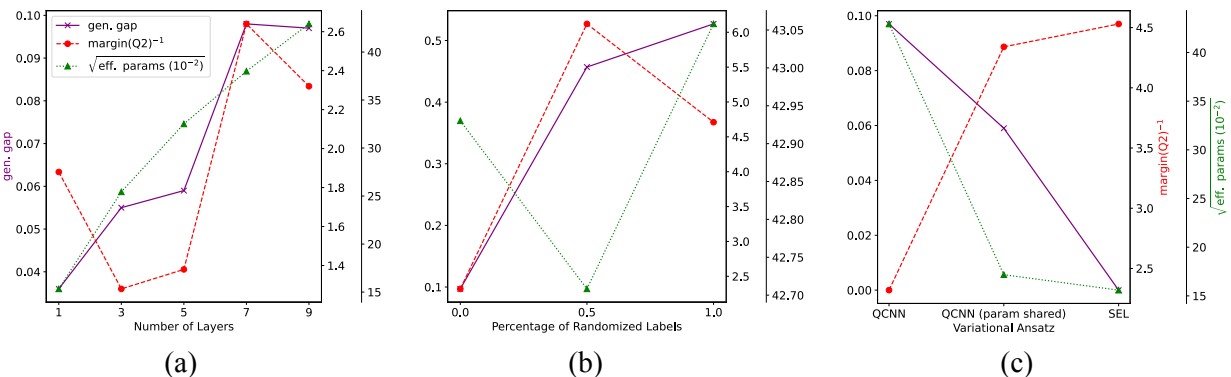

*Figure 6.* Reproduction of Figure 2 with 10-qubit generalized cluster Hamiltonian.

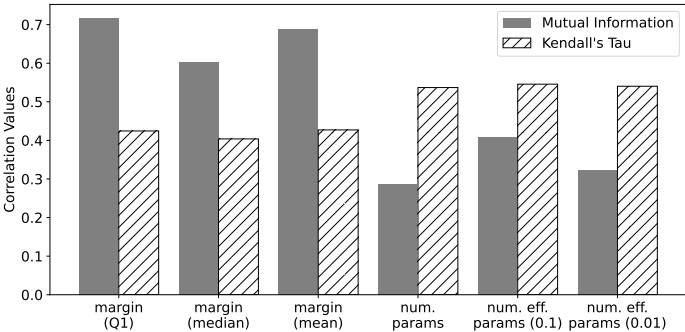

*Figure 7.* Reproduction of Figure 3 for 10-qubit generalized cluster Hamiltonian.

8-QUBIT XXZ-HEISENBERG SPIN CHAIN

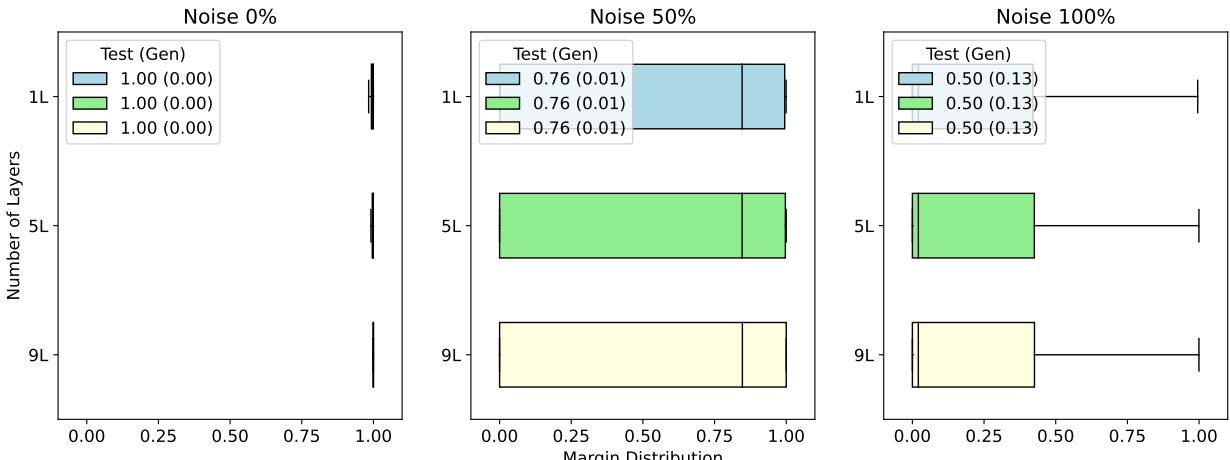

*Figure 8.* Reproduction of Figure 1 with 8-qubit XXZ-Heisenberg spin chain.

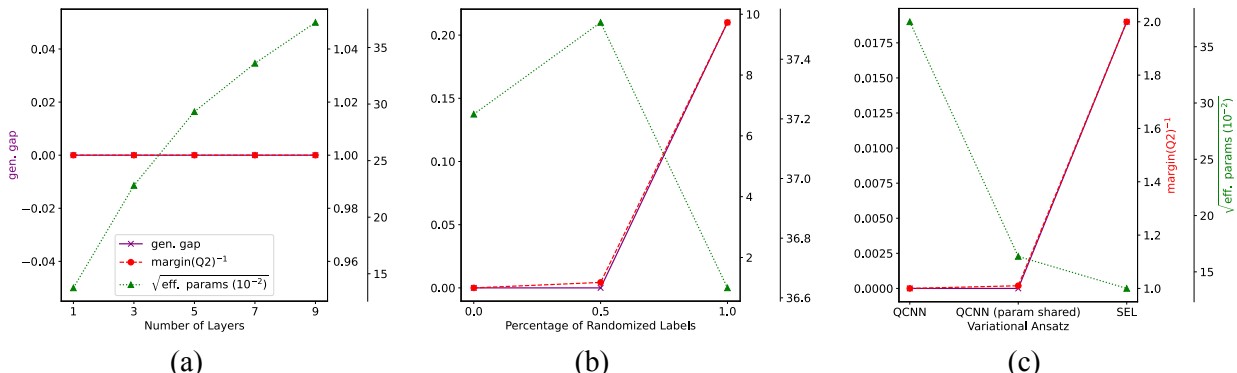

*Figure 9.* Reproduction of Figure 2 with 8-qubit XXZ-Heisenberg spin chain.

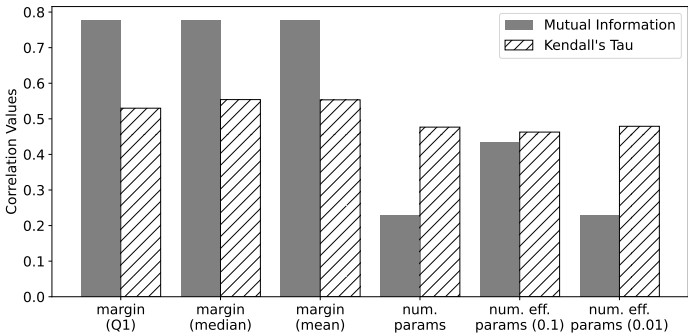

*Figure 10.* Reproduction of Figure 3 with 8-qubit XXZ-Heisenberg spin chain.

10-QUBIT XXZ-HEISENBERG SPIN CHAIN

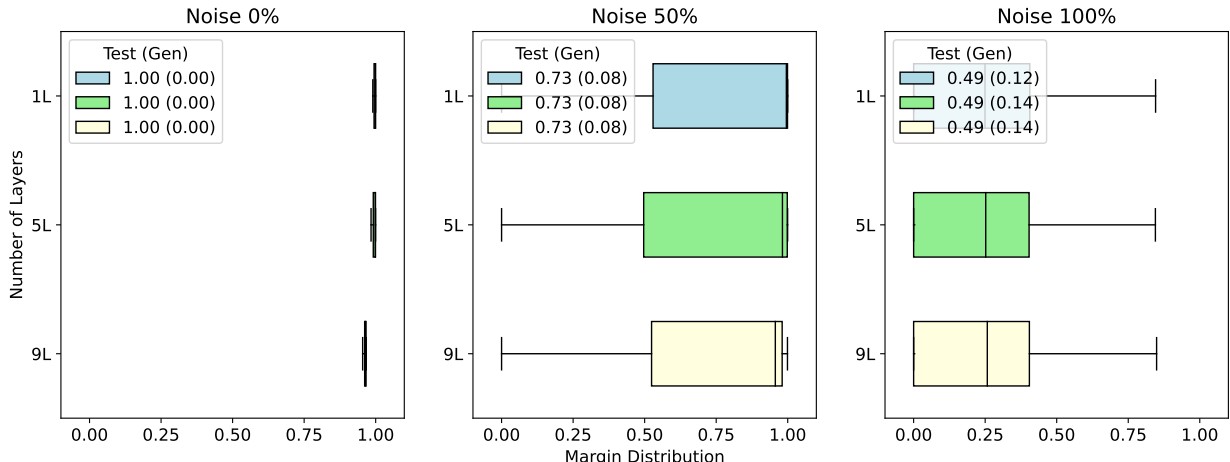

*Figure 11.* Reproduction of Figure 1 with 10-qubit XXZ-Heisenberg spin chain.

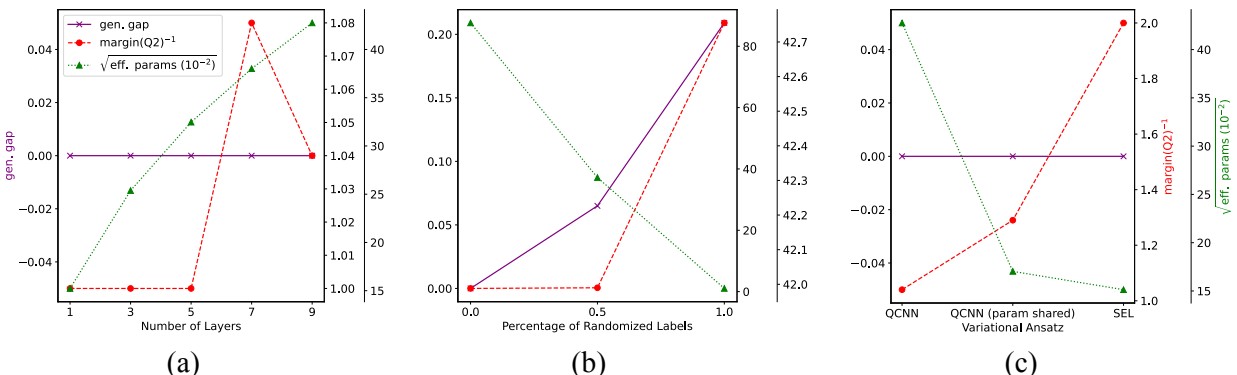

*Figure 12.* Reproduction of Figure 2 with 10-qubit XXZ-Heisenberg spin chain.

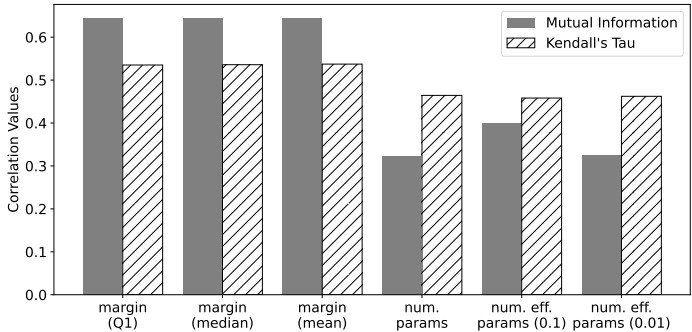

*Figure 13.* Reproduction of Figure 3 with 10-qubit XXZ-Heisenberg spin chain.

8-QUBIT TRANSVERSE FIELD ISING MODEL

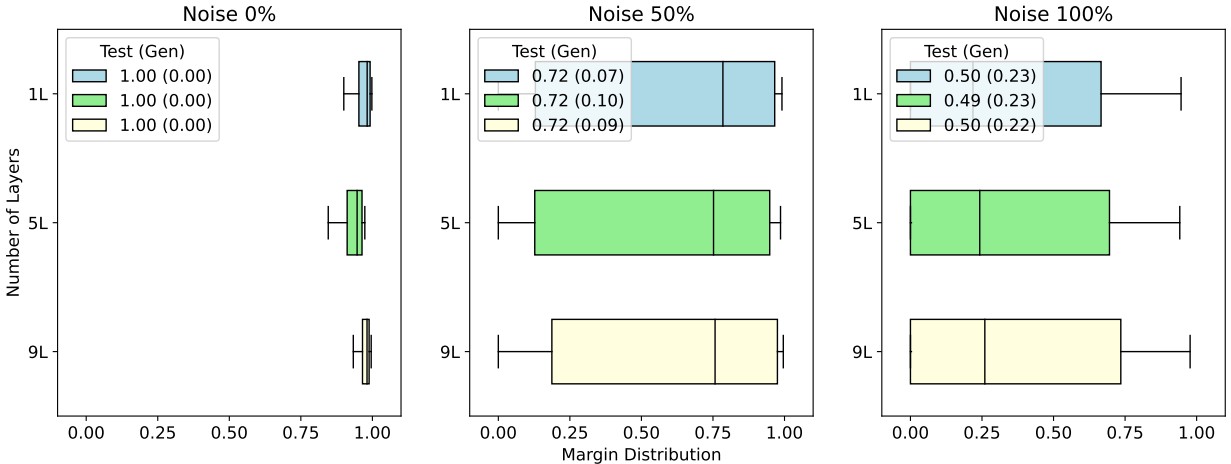

*Figure 14.* Reproduction of Figure 1 with 8-qubit Transverse Field Ising Model.

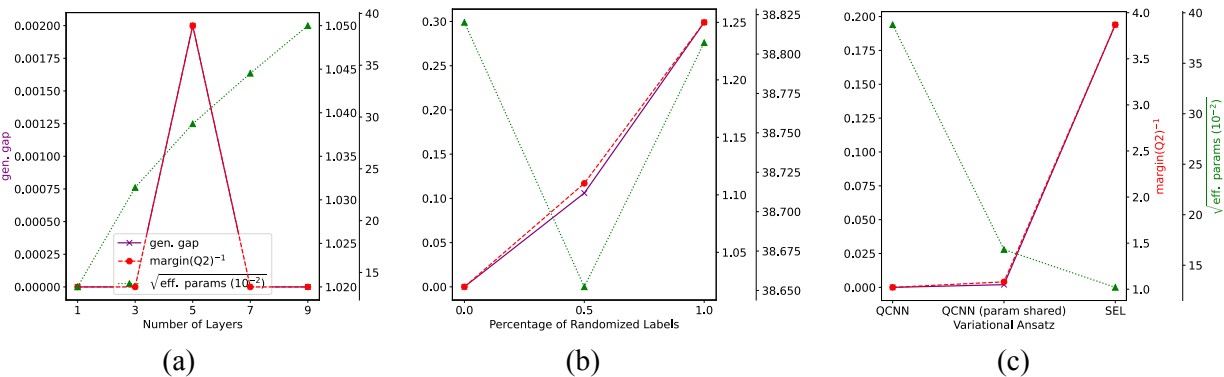

*Figure 15.* Reproduction of Figure 2 with 8-qubit Transverse Field Ising Model.

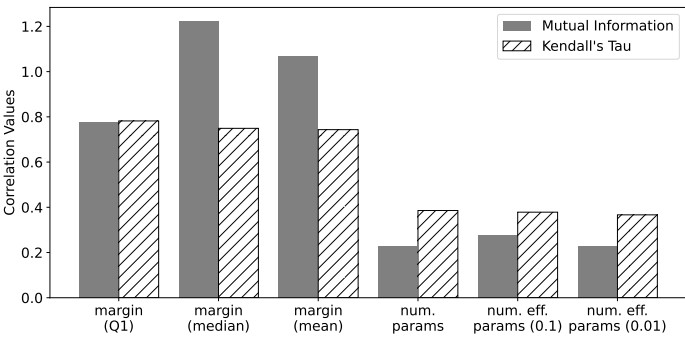

*Figure 16.* Reproduction of Figure 3 with 8-qubit Transverse Field Ising Model.

10-QUBIT TRANSVERSE FIELD ISING MODEL

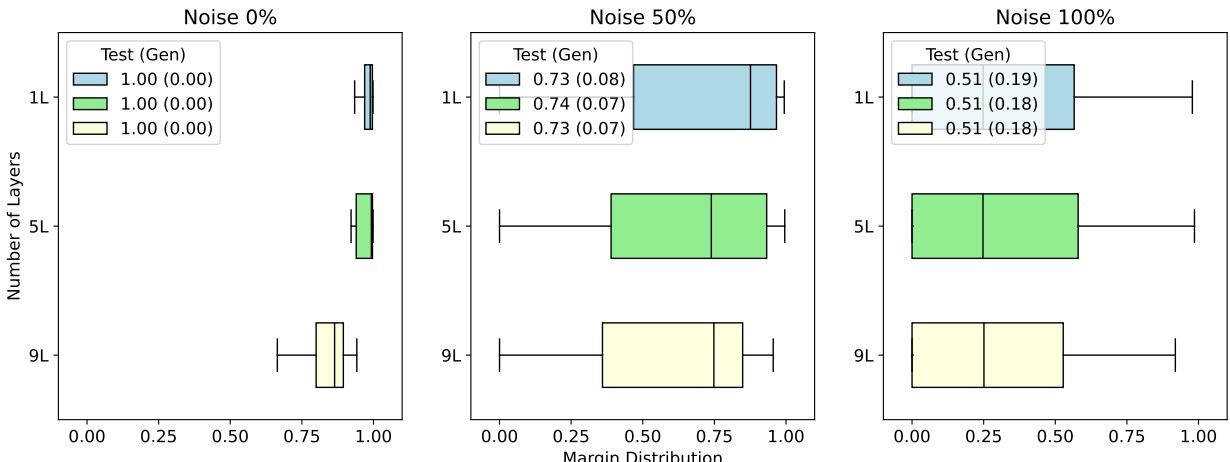

*Figure 17.* Reproduction of Figure 1 with 10-qubit Transverse Field Ising Model.

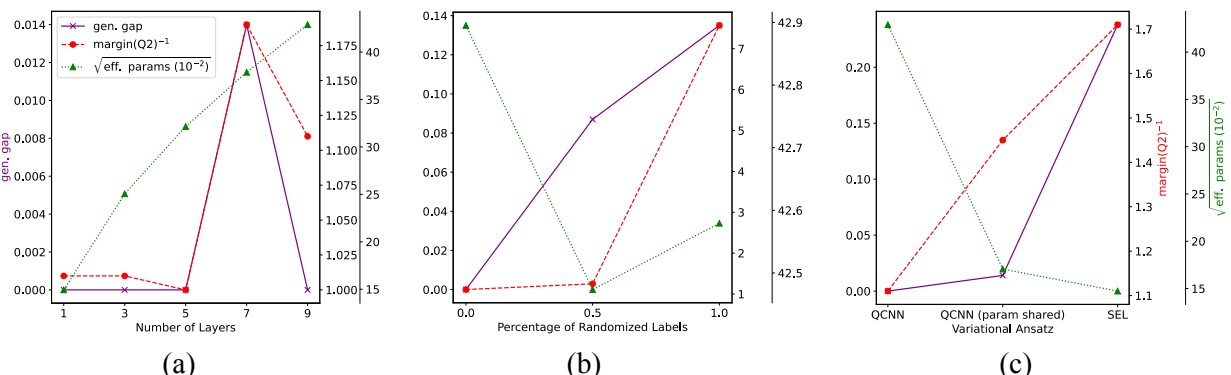

*Figure 18.* Reproduction of Figure 2 with 10-qubit Transverse Field Ising Model.

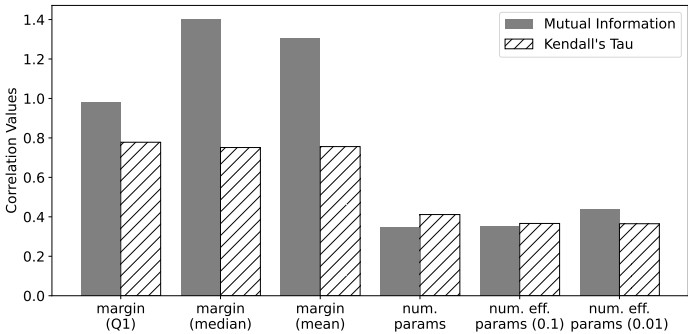

*Figure 19.* Reproduction of Figure 3 with 10-qubit Transverse Field Ising Model.

MARGIN DISTRIBUTION PLOTS

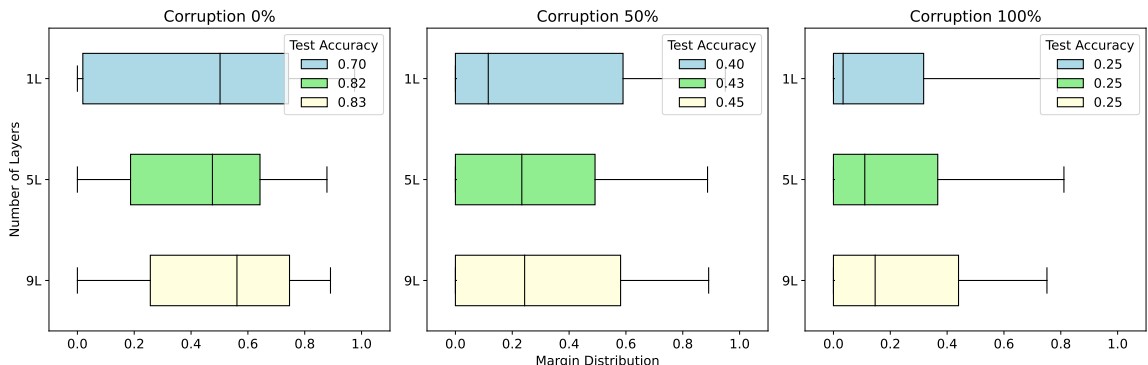

(a) QCNN with shared parameters

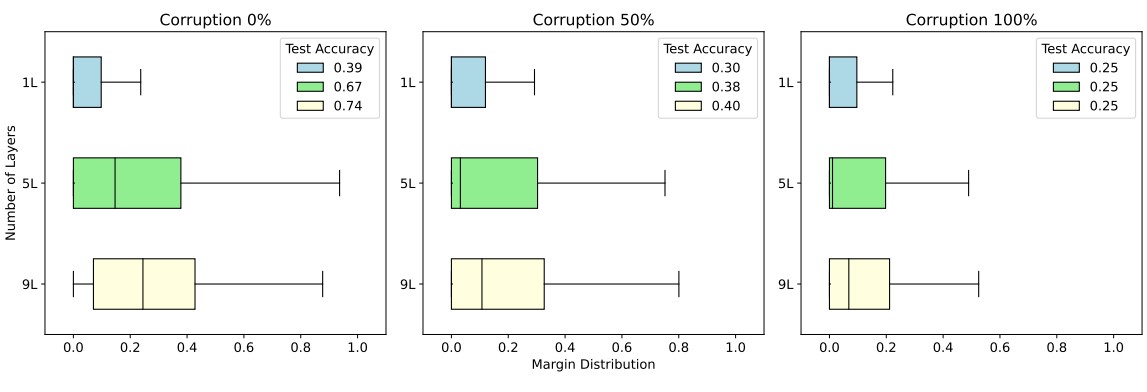

(b) Strongly Entangling Layers

*Figure 20.* Reproduction of margin distribution (Figure 1) using different variational ansätze.

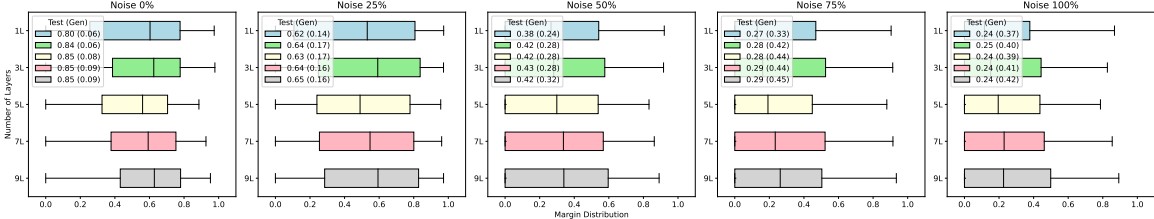

*Figure 21.* Fine-grained margin distribution plots.

Figure 1 presents results for QCNNs, where the two-qubit PQCs are permitted to have distinct parameter values within the convolutional layer. In Figure 20, we provide additional experimental results using different variational ansätze: QCNN with shared parameters and Strongly Entangling Layers. Strongly Entangling Layers consist of repeated single-qubit rotations and two-qubit CNOT gates, provided as a built-in feature by PennyLane (Bergholm et al., 2020). Consistent with the findings from the main text, we observe a decline in test accuracy as the noise level rises, accompanied by a left-skewed margin distribution. Additionally, Figure 21 provides plot with more fine-grained experimental results. These details were excluded from the main text purely for reason of display conciseness. The figure show performance across corruption levels of $0\%, 25\%, 50\%, 75\%, 100\%$ as well as for $1, 3, 5, 7, 9$ numbers of layers.

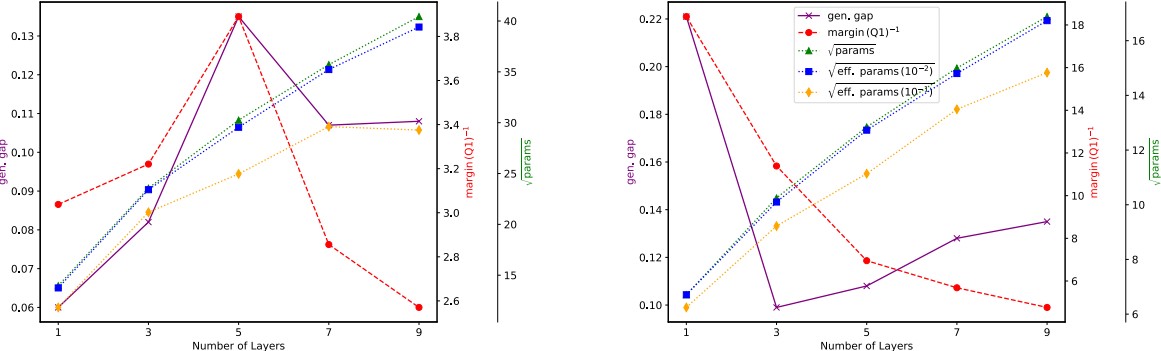

(a) Lower quartile of margin versus parameter-based metrics for predicting the generalization gap.

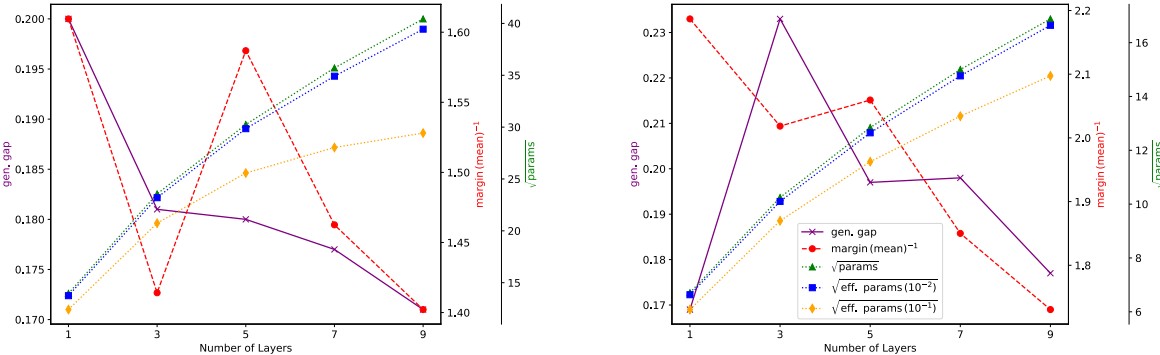

(b) Margin mean versus parameter-based metrics for predicting the generalization gap.

*Figure 22.* Comparison of the lower quartile (a) and the mean (b) of the margin against parameter-based metrics for predicting the generalization gap. The experiments use QCNNs (left) and QCNNs without parameter sharing (right) as the variational ansatz.

Figure 22 (a) and (b) compare parameter-based metrics with the lower quartile and mean of the margin, respectively. Consistent with the results in the main text, the margin-based metrics more effectively capture the generalization gap across all settings.

### A.4. Margin mean and Quantum State Discrimination

In Section 5, we established a connection between quantum state discrimination and margin. Specifically, we showed that the margin mean is upper bounded by the trace distance between quantum state ensembles, making a large initial trace distance essential for achieving strong generalization. Here, we provide the detailed derivations.

Equation 2 demonstrated that margin mean can be expressed as:

$$\bar{\mu} = 2\mathrm{Tr}(p^+\rho^+ E^*_{+1}) + 2\mathrm{Tr}(p^-\rho^- E^*_{-1}) - 1. \tag{15}$$

Recall that $\{E^*_{\pm 1}\}$ forms a set of POVMs, meaning $E^*_{+1} + E^*_{-1} = I$. Utilizing this property, we can rewrite:

$$
\begin{aligned}
\mathrm{Tr}(p^+\rho^+ E^*_{+1}) + \mathrm{Tr}(p^-\rho^- E^*_{-1}) &= p^- + \mathrm{Tr}\left((p^+\rho^+ - p^-\rho^-)E^*_{+1}\right) \\
&= p^+ - \mathrm{Tr}\left((p^+\rho^+ - p^-\rho^-)E^*_{-1}\right) \\
&= \frac{1}{2} + \frac{1}{2}\mathrm{Tr}\left((p^+\rho^+ - p^-\rho^-)(E^*_{+1} - E^*_{-1})\right),
\end{aligned}
\tag{16}
$$

where the final equality averages the first two.

We are particularly interested in the maximum of $\text{Tr}(p^+\rho^+ E^*_{+1}) + \text{Tr}(p^-\rho^- E^*_{-1})$. Consider the spectral decomposition $p^+\rho^+ - p^-\rho^- = \sum \lambda_i |\psi_i\rangle\langle\psi_i|$, where $\lambda_i$ are real due to Hermitian nature of the density matrix. The maximum occurs when $E^*_{\pm 1}$ are *Helstrom measurements*, which are projects onto the positive and negative subspaces, i.e., $E^*_{+1} = \sum_{\{i:\lambda_i>0\}} |\psi_i\rangle\langle\psi_i|$ and $E^*_{-1} = \sum_{\{i:\lambda_i<0\}} |\psi_i\rangle\langle\psi_i|$.

Thus, we have:

$$
\begin{aligned}
\max_{E^*_{\pm 1}} \text{Tr}(p^+\rho^+ E^*_{+1}) + \text{Tr}(p^-\rho^- E^*_{-1}) &= \frac{1}{2} + \frac{1}{2}\sum_i |\lambda_i| \\
&= \frac{1}{2} + \frac{1}{2}D_{\text{tr}}(p^+\rho^+, p^-\rho^-).
\end{aligned}
\tag{17}
$$

Substituting this into Equation 15, we obtain $\bar{\mu} \leq D_{\text{tr}}(p^+\rho^+, p^-\rho^-)$, where equality holds when $E^*_{\pm 1}$ are the Helstrom measurements.

