# OpenReview forum: "Understanding Generalization in Quantum Machine Learning with Margins"
_ICML.cc/2025/Conference — ICML 2025 poster_

### Official Review · Reviewer_5bgA · 2025-03-12

**Overall Recommendation:** 3

**Summary:**

The authors address generalization in quantum machine learning by introducing a margin-based framework. The authors critique traditional uniform generalization bounds, which have been shown to be ineffective in both classical and quantum settings, and propose margin-based generalization bounds as a more reliable alternative. They extend classical margin-based theory to quantum neural networks by leveraging Lipschitz continuity and matrix covering techniques. Their experiments on quantum phase recognition datasets demonstrate a strong correlation between margin distribution and generalization performance, surpassing traditional metrics like parameter count. They also establish a connection between margins and quantum state discrimination, highlighting how maximizing the separability of quantum embeddings improves generalization. Their findings suggest that margin-based methods provide better theoretical insights and practical guidance for designing QML models with improved generalization capabilities.

**Claims And Evidence:**

The claims made in the submission are supported by clear and convincing evidence

**Essential References Not Discussed:**

No

**Experimental Designs Or Analyses:**

The optimized 8-qubit QCNNs and the generalization gap analysis are well-designed experiments. However, the discretization of the experiments is somewhat coarse, which may impact the precision of the results.

**Methods And Evaluation Criteria:**

The proposed methods and evaluation criteria make sense for the problem, but they can be improved with more datasets.

**Other Comments Or Suggestions:**

1. Add a relevant work section

**Other Strengths And Weaknesses:**

Strengths:

1. The method is well-motivated, addressing the limited understanding of generalization in quantum machine learning. The connection between quantum state discrimination and generalization is a novel and valuable contribution to QML theory.

2. The paper is well-written, and the results are engaging and clearly presented.

3. The method outperforms the compared approaches, particularly in the comparative analysis. The experiments show a strong correlation between margin distribution and generalization performance, supporting the theoretical claims.

4. The application of QML to classical data using quantum embeddings is not novel but is a useful and relevant demonstration.

Weaknesses:

1. Limited experiments, broader testing across different parameters is needed for statistically robust validation.

2. Generalization gap results are mixed, especially in the appendix.

3. Lacks comparison with other  methods, relying mainly on comparison with effective parameter analysis.

**Questions For Authors:**

1, How would the method of margins compare to other methods in terms of robustness, or how generalizable is this approach.

## Update after Rebuttal

The additional results and clarifications strengthen the massage of the already interesting work. However, the new experiments are similar to the existing ones and the sections need to be improved to match the style of ICML better, e.g. clear Relevant works and limitations are missing. Additionally, I am skeptical about the significance of the results, partly due to the limited experimentation, in regard to a venue like ICML.

**Relation To Broader Scientific Literature:**

The authors relate to the previous margin-based generalization. The paper is missing a related works section.

**Theoretical Claims:**

No proofs in the main manuscript.

---

> ### Author Rebuttal · Authors · 2025-04-01
>
> We sincerely thank Reviewer 5bgA for insightful comments. Below, we address each point directly and explain how we plan to incorporate your suggestions.
>
> ---
> ### 1) **Limited experiments; broader testing is needed for statistically robust validation**
> To strengthen empirical validation, we have now conducted extensive additional experiments on two canonical quantum many-body datasets: **Transverse Field Ising Model** and **XXZ-Heisenberg chain**. We repeated all main-text experiments on these datasets; results are summarized clearly at the following anonymous links: [TFIM](https://tinyurl.com/47mpp43c) and [XXZ-Heisenberg](https://tinyurl.com/3r598nmm).
> For both TFIM and XXZ datasets, the margin consistently correlate with generalization performance. This strongly reinforces our central claims regarding the effectiveness of margin bounds for QNNs. These additional datasets significantly broaden our experimental scope, providing comprehensive validation of margin-based generalization theory.
>
> We appreciate the reviewer highlighting the coarse discretization of our initial experimental results (originally QCNN layers = $\{1,5,9\}$). To comprehensively validate our claims, we now provide fine-grained experimental results covering $\{1,3,5,7,9\}$ layers. We also extend these detailed margin distribution analyses to the TFIM and XXZ-Heisenberg model. Results are summarized [at this anonymous link](https://tinyurl.com/5n6j7us3). We will explicitly incorporate these detailed results into the appendix of our revised manuscript.
>
> ---
> ### 2) **Generalization gap results are mixed, especially in the appendix**
> We acknowledge the reviewer’s observation regarding mixed generalization gap results presented in the appendix. However, even within these initial results, margin-based metrics (both Q1 and mean) still consistently outperform all parameter-based metrics.
>
> To further address this concern, we conducted additional experiments using the TFIM and XXZ-Heisenberg datasets, available [at this anonymous link](https://tinyurl.com/mrx8943h). These new results demonstrate highly stable and consistent correlations between margin-based metrics (Q1 and mean) and the generalization gap, without mixed outcomes. This further supports our central thesis that margin-based metrics offer stable and robust predictors of generalization in QNNs.
>
> ---
> ### 3) **Lack of comparison with methods beyond parameter-based metrics**
> We appreciate the reviewer highlighting this point. Compared to classical machine learning, Quantum Machine Learning (QML) is a much newer field, and currently has relatively few established methods for evaluating generalization. While several recent works have proposed bounds from different theoretical perspectives, most ultimately fall within the class of parameter-based or uniform generalization bounds.
>
> For instance:
>
> - [Banchi et al.](https://tinyurl.com/yjx386x9) derive information-theoretic bounds based on the mutual information between the training data and the parametric quantum states. The analysis is based on Rademacher complexity, reflecting limitations shared with other uniform bounds, such as that proposed by [Caro et al](https://tinyurl.com/mrsrrjk4).
>
> - [Abbas et al.](https://tinyurl.com/mw2ua99y) propose using the effective dimension, defined via the Fisher information matrix, to capture model complexity. This approach captures model expressivity through parameter sensitivity and thus also remains fundamentally parameter-based.
>
> Given this, we chose to compare against more recent work by [Caro et al.](https://tinyurl.com/mrsrrjk4) as a representative of this broader class of approaches. Their bounds are constructed using covering numbers and Rademacher complexity, directly in terms of trainable parameters and model capacity. This makes their work both relevant and representative of the prevailing generalization bounds in QML to date.
>
> Moreover, the recent critical analysis by [Gil-Fuster et al.](https://tinyurl.com/4kjnvb44) directly points to inherent shortcomings of parameter-based uniform bounds, motivating us to benchmark against parameter-based methods explicitly.
> We will clearly outline these justifications in our revised manuscript, contextualizing our choice of comparisons and the current state of generalization theory in QML.
>
> ---
> ### **Additional Points**
>
> **Related Work Section:**  We agree and thank the reviewer for this suggestion. We will include a concise **Related Work** section in the revised manuscript, situating our contributions within recent quantum generalization literature.
>
> **Proofs in the Main Manuscript:**  To maintain readability, detailed proofs were provided in Appendix A.1. We will refer readers to these proofs from the main text, clearly guiding navigation without compromising readability.
>
> ---
> We greatly appreciate Reviewer 5bgA’s valuable suggestions and believe these adjustments significantly enhance the clarity, robustness, and overall quality of our manuscript.

---

> > ### Comment · Reviewer_5bgA · 2025-04-02
> >
> > The paper is interesting and presents novel aspects, but I still find the experimentation somewhat limited. I will maintain a skeptical 3.

---

> > > ### Author Response · Authors · 2025-04-09
> > >
> > > We sincerely thank Reviewer 5bgA for highlighting the concern regarding the scope and precision of our experiments. Here, we would like to emphasize additional experiments we performed in our initial rebuttal and present further experiments.
> > >
> > > ---
> > >
> > > ## **1. Additional Dataset**
> > >
> > > In our original rebuttal, we expanded our empirical analysis by introducing two widely recognized quantum many-body benchmarks datasets: the Transverse Field Ising Model (TFIM) and the XXZ-Heisenberg model. We systematically reproduced all main-text experiments using these additional datasets, ensuring broad applicability and robustness of our margin-based generalization theory. Detailed results can be accessed via the following links:
> > >
> > > - [Github Link for TFIM experiments](https://tinyurl.com/47mpp43c)
> > > - [Github Link for XXZ experiments](https://tinyurl.com/3r598nmm)
> > >
> > > ## **2. Fine-Grained Experiments: QCNN Layers**
> > > To address your initial concern regarding coarse experimental discretization, we refined our experimental design by expanding the number of QCNN layers studied. Originally, our manuscript included results for QCNN layers $1, 5, 9$. In our initial rebuttal, we enhanced this by incorporating layers $1, 3, 5, 7, 9$. We repeated these fine-grained analyses on both the TFIM and XXZ datasets, demonstrating the robustness and consistency of our results. Results can be found at:
> > > - [Github Link for Fine-Grained Experiments: Number of QCNN Layers](https://tinyurl.com/5n6j7us3)
> > >
> > > ## **3. Fine-Grained Experiments: Noise Levels**
> > > While our initial rebuttal focused on adding more datasets and refining QCNN depth, we have conducted additional experiments in this second round to further broaden the scope of experimental testing, in response to the continued concern regarding limited experimental scope. Specifically, we enhanced our study of label noise by performing more fine-grained experiments across five levels of label randomization: 0%, 25%, 50%, 75%, 100%. This extends our original setup, which included 0%, 50%, 100%, and provides greater experimental granularity.
> > >
> > > These newly added experiments were repeated on both the TFIM and XXZ datasets to confirm the consistency and robustness of the observed trends across different parameter settings and data distributions. Detailed results are available here:
> > > - [Github Link for Fine-Grained Experiments: Noise Levels](https://tinyurl.com/3s4xtu5j)
> > >
> > > ---
> > > ## **Comparative Analysis with Existing QML Literature**
> > >
> > > To illustrate the comprehensiveness and depth of our experiments compared to the most well-known and foundational results in generalization in Quantum Machine Learning (QML), we summarize key aspects of prior works in the following table:
> > >
> > >
> > > | Work             | Datasets Used                                                   | Variational Ansatz                                              | Number of Layers           | Label Randomization                         |
> > > |------------------|-----------------------------------------------------------------|-----------------------------------------------------------------|----------------------------|---------------------------------------------|
> > > | [Abbas et al.](https://tinyurl.com/mw2ua99y)     | 1 classical (Iris)                                              | 1 (Strongly Entangling Layers)                                  | 1 (fixed)                  | N/A                                         |
> > > |  [Caro et al.](https://tinyurl.com/mrsrrjk4)      | 1 quantum (cluster)                                | 1 (QCNN)                                                        | 1 (fixed)                  | N/A                                         |
> > > | [Banchi et al.](https://tinyurl.com/yjx386x9)    | 1 quantum (TFIM), 1 classical (2-moon)                          | 2 (Fidelity Classifier (TFIM), Single-qubit data-reuploading (2-moon))   | 1 (fixed)                      | N/A                                         |
> > > | **Ours**         | 3 quantum (cluster, TFIM, XXZ), 3 classical (MNIST, Fashion-MNIST, Kuzushiji-MNIST) | 3 (QCNN, parameter sharing QCNN, Strongly Entangling Layers) | 5 (1, 3, 5, 7, 9 layers)   | 5 levels (0%, 25%, 50%, 75%, 100%)          |
> > >
> > > This comparison illustrates that our experimental setup is designed to provide broader empirical coverage—spanning multiple datasets, architectures, and levels of label randomization—extending beyond the experimental analyses conducted in prior state-of-the-art studies.
> > >
> > > Finally, we would like to note that, following the anonymous review process, we plan to release our code and experimental data on a public repository, to support reproducibility and facilitate further exploration by the community.
> > >
> > > We hope these additional clarifications and comprehensive experimental refinements fully address your concerns.

---

### Official Review · Reviewer_rwMH · 2025-03-13

**Overall Recommendation:** 3

**Summary:**

This paper establish a margin-based generalization bound for multiclass classification with Quantum Neural Networks, adapting techniques from classical neural networks to the quantum domain. This approach interprets quantum measurements as nonlinear activations and extends matrix covering techniques to complex-valued spaces. Through experiments on quantum phase recognition datasets, they demonstrate that margin-based metrics strongly correlate with generalization performance, even when traditional metrics like parameter count fail. Also they conduct experiments on three quantum embedding methods and showing that Neural Quantum Embedding (NQE), a classical-quantum hybrid approach, enhances generalization by yielding larger margins through increased data distinguishability.

**Claims And Evidence:**

The claims made in the paper are generally well-supported by evidence through both theoretical development and experimental validation: The margin-based generalization bound is mathematically derived with clear steps, extending established techniques from classical to quantum settings. Experimental evidence strongly supports the predictive power of margin-based metrics for generalization. These evidences include usage of multiple datasets, testing various models and challenging scenarios like randomized labels

**Essential References Not Discussed:**

No to the best of my knowledge.

**Experimental Designs Or Analyses:**

The experimental designs and analyses in the paper are generally sound: Multiple experimental setups testing different aspects of the theory (generalization gap prediction, margin distribution analysis, quantum embedding comparison), Appropriate statistical measures (mutual information, Kendall rank correlation) to quantify relationships, Controls for variability by averaging results over 15 repetitions with different training samples, Systematic exploration of hyperparameters (layers, noise levels, embedding strategies)

**Methods And Evaluation Criteria:**

The proposed methods and evaluation criteria are appropriate for studying generalization in quantum machine learning. They are using margin-based approach is sensible given its success in classical deep learning and the demonstrated limitations of uniform bounds. Also the theoretical framework acknowledges quantum-specific properties like POVM measurements. For the evaluation, they test on both quantum data and classical data to show broader applicability. Comparing against parameter-based metrics establishes relative improvement over existing approaches.

**Other Comments Or Suggestions:**

It would be good to include a diagram illustrating the conceptual relationship between quantum embeddings, trace distance, and margins. Figure labels in Figure 2 are somewhat difficult to understand.

**Other Strengths And Weaknesses:**

Strengths: It successfully bridges classical margin theory with quantum information theory, creating a unified framework for understanding QML generalization. It provides actionable guidance for designing better quantum embeddings based on trace distance maximization. It tests across multiple datasets, model architectures, and hyperparameters. Mathematical derivations are well-structured and the experimental results are clearly presented.

Weaknesses: Experiments on small qubit systems (8 qubits) may not fully represent challenges of larger quantum systems. No discussion of computational costs for calculating margins versus parameter counts. Lacks analysis of how real quantum hardware noise might affect the margin-based approach.

**Questions For Authors:**

How would your margin-based approach scale to larger quantum systems (beyond 8 qubits)? Do you anticipate any theoretical or practical challenges?
What is the computational overhead of calculating margin-based metrics compared to parameter-based metrics?
How might real quantum hardware noise affect your margin-based approach to generalization?
Your small training set (20 samples) for QPR experiments raises questions about statistical significance. Have you verified these results hold with larger datasets?

**Relation To Broader Scientific Literature:**

This paper brings margin-based framework to the quantum machine learning area, continuing the trend of non-uniform generalization bounds that better predict real performance. Also it links fundamental quantum information theory (trace distance, Helstrom measurements) to machine learning performance, connecting QML to established quantum information concepts.

**Theoretical Claims:**

There are no significant issues in the theoretical proofs to the best of my knowledge.

---

> ### Author Rebuttal · Authors · 2025-04-01
>
> We thank Reviewer rwMH for thoughtful evaluation.
>
> ---
> ### 1) **Experiments on small qubit systems**
> We agree that our experiments were conducted on relatively small quantum systems (8-qubit QCNNs). While it is possible to increase the number of qubits by one or two, we chose 8 qubits as a practical design decision. First, powers of two are a natural choice for QCNNs due to their hierarchical structure. Second, we do not expect a small increase in the number of qubits to significantly affect the generalization behavior.
>
> While extending the numerical studies to much larger systems (16 or 32 qubits) would be of interest, classical simulation becomes exponentially expensive with the number of qubits. For example, training a single 8-qubit QCNN requires ~**0.5hours** on state-of-the-art commercial CPUs. Our experiments were performed across extensive hyperparameters: [1,3,5,7,9] for QCNN layers, [0, 0.5, 1] for label noise, [QCNN, QCNN_shared, SEL] for variational ansatz. Each setting was repeated **15 times** to control variability, resulting in a total time of ~**300 hours**. Increasing the number of qubits by one would double simulation time, quickly becoming computationally impractical.
>
> Nonetheless, our theoretical results are valid beyond 8-qubits. Our margin bound scales linearly with the number of qubits, suggesting that larger systems would exhibit analogous behaviors, provided the sample grows proportionally. Thus, while explicit numerical validation at larger scales is limited by classical computational constraints, our theory remains scalable.
>
> Moreover, [prior work](https://tinyurl.com/4kjnvb44) has shown that QNNs can efficiently memorize polynomially increasing training data, implying uniform bounds can remain vacuous even in larger quantum systems. Thus, the generalization challenges we address are fundamental to QML, not artifacts of small system size. This further motivates non-uniform measures, such as our proposed margin bound, which remain meaningful regardless of scale.
>
> ---
> ### 2) **Computational costs for calculating margins vs. parameter counts**
> The margin for a single data point $(\rho, y)$ is defined as $h(\rho)_y - \max\_{j \neq y} h(\rho)_j$. In principle, evaluating margin distributions for an $m$-sample dataset requires $O(m)$ quantum circuit executions. However, practically, the predictions $h(\rho)$ are computed as part of the final step of model training. Therefore, calculating margin distributions **requires no additional computational costs** beyond what is already used during model training. We will explicitly mention this practical consideration in our revised manuscript.
>
> ---
> ### 3) **Lack of analysis regarding real quantum hardware noise**
> Our work offers a new perspective on how quantum noise affects generalization by connecting margin bounds to quantum state discrimination. Quantum noise, which are contractive CPTP maps ($\Lambda$), reduces trace distance $D(\Lambda(\rho_1),\Lambda(\rho_2)) \leq D(\rho_1,\rho_2)$. In Section 4, we explicitly show that the margin mean is upper-bounded by the trace distance. Consequently, quantum noise shrinks trace distance and shifts margin distributions leftward, resulting in decreased generalization performance.
>
> This connection is not captured by previous generalization bounds in QML and, represents a meaningful advancement in the theoretical understanding of generalization. The margin-based generalization framework naturally incorporates quantum noise effects, making it uniquely insightful. We agree that deeper experimental investigation of noise effects would be a valuable next step, and we believe our work provides a strong theoretical foundation for such studies.
>
> ---
> ### 4) **Small training set (20 samples) and statistical significance**
>
> We recognize that our QPR experiments employed relatively small training datasets (20 samples). This experimental choice was deliberate and aligned precisely with previously established works ([Caro et al.](https://tinyurl.com/mrsrrjk4) and [Gil-Fuster et al.](https://tinyurl.com/4kjnvb44)), which served as direct references. Specifically, Gil-Fuster et al. experimentally demonstrated that QCNNs could overfit randomized labels with small datasets, questioning the effectiveness of uniform generalization bounds presented in Caro et al. Since a core contribution of our work is to propose margin bounds as a tight, non-uniform alternative to these uniform bounds, we intentionally retained similar experimental setups for direct comparison.
>
> Furthermore, to bolster empirical validation, our manuscript also presents additional extensive experiments on classical datasets (MNIST, Fashion-MNIST, and Kuzushiji-MNIST) in Section 4, each consisting of approximately **12,000 samples**. In this large-sample regime, we consistently observed strong correlations between margin metrics and generalization, confirming the statistical significance and robustness of our claims beyond the small-sample scenario.

---

> > ### Comment · Reviewer_rwMH · 2025-04-08
> >
> > Thanks authors for the detailed response. I especially thanks for the explanation of 2), its pretty clear. Also for 3), it could be really exciting to see more results that consider the noise - however, I also understand it could be even more expensive for your simulation. Regarding your 1), 8 qubits for 0.5 hrs for a QCNN probably means you are not using advanced simulator, you can try with stabilizer tensor network simulator, I assume it could enable your simulation for ~10-12 qubits system. Since its 2025, 8 qubits small scale experiment is somehow not acceptable to me, but I do agree with the novelty of the paper, I would like to stand with positive score.

---

> > > ### Author Response · Authors · 2025-04-09
> > >
> > > We sincerely thank Reviewer rwMH again for your detailed and insightful engagement with our work. We're delighted to hear that our clarification of points (2) and (3) was helpful and clear. Following your advice, we are currently preparing additional experiments on larger quantum systems (10–12 qubits), and we plan to include these results in the revised version of the manuscript, before camera-ready. We greatly value your input and thank you again for your constructive review.

---

### Official Review · Reviewer_uadb · 2025-03-13

**Overall Recommendation:** 3

**Summary:**

The manuscript describes a theoretical and experimental analysis of quantum machine learning models, with the focus on generalization bounds. The authors build on prior quantum machine learning results indicating vacuity of bounds based on parameter count or other measures of complexity of the hypothesis space. They adapt bounds based on margin distribution from classical to quantum ML, and show that it explains generalization gap more accurately than parameter-based metrics.

**Claims And Evidence:**

The two main claims are:

1) Generalization in QML theoretically depends on margin. Here, the authors adapt existing margin-based generalization bound to take into account the characteristics of quantum models. The evidence, in the form of proof of the bound, is sound. The assumptions relating to the quantum nature of the model architecture (model is a parameterized unitary followed by measurement, projective or more generally, POVM) are also sound.

2) The bound from 1) has practical relevance. Here, the authors perform an experimental analysis evaluating to what extent inverse margin aligns with generalization gap. The experimental evidence is convincing (inverse margin aligns well, qualitatively in Fig 2 and quantitatively in Fig 3) with generalization gap for varying number of layers in the model, varying internal architecture, and varying randomization of the dataset). The evidence is, however, only limited to one dataset.

The two results above are focused on quantum model working on quantum input data. The paper also extends the margin-based generalization analysis to classical data – quantum model setup, and provides a margin-based explanation of the previously observed relationship between the choice of how classical data is embedded into quantum state and generalization. The theoretical link between mean margin and quantities previously shown to lower-bound loss is sound, and the authors show experimental evidence for the link between margin and test accuracy. The experimental limited (three MNIST-based datasets, one model QCNN). The presentation in the main manuscript is in terms of test accuracy, without showing training set accuracy, which does not directly support the claims that are related to generalization gap. Some limited exploration of generalization gap is provided in the Appendix (A.2, Fig. 6), but lacks details in the description (e.g. which MNIST dataset is used in Fig. 6?).

**Essential References Not Discussed:**

None noted.

**Experimental Designs Or Analyses:**

The overall choice of experimental setup (models and dataset in Sect. 3 & 4) is sound, though limited in breadth.

**Methods And Evaluation Criteria:**

Empirical results are somewhat limited in scope, and (in section 4) not perfectly aligned with theory (see above for details).

**Other Comments Or Suggestions:**

The authors should consider moving the generalization gap-focused results into the main manuscript (Fig. 6), providing more detailed description.

**Other Strengths And Weaknesses:**

In terms of strengths, the paper extends our understanding of generalization in quantum machine learning.
One weakness of the work is the limited nature of the experimental evidence. Another weakness is the separate treatment of classical data scenario (Sec. 4) from the earlier parts (Sect. 2). The paper would be stronger if it could present a unifying theoretical framework that incorporates properties of the embedding circuit as parameters in the bounds, in the same manner as different options for measurements (properties of E) are incorporated.

**Questions For Authors:**

None.

**Relation To Broader Scientific Literature:**

The paper extends prior findings on generalization bounds for quantum data-quantum model, and prior findings about quantities that affect how quantum embeddings related to effectiveness of classical data – quantum model; in both cases they extend the prior findings by introducing margin as an explanatory variable. This mirrors earlier developments in classical machine learning.

**Theoretical Claims:**

Correctness of proofs in Appendix has not been checked in detail.

---

> ### Author Rebuttal · Authors · 2025-03-31
>
> We thank Reviewer uadb for thorough evaluation and insightful suggestions.
>
> ---
> ### 1) **Experimental Scope and Additional Results**
> We acknowledge the reviewer's concern regarding the breadth of empirical evidence. To address this directly, we conducted extensive additional experiments on two canonical quantum many-body benchmarks: **Transverse Field Ising Model** and **XXZ-Heisenberg Model**.
>
> These additions go beyond the original QPR dataset and validate that our margin-based framework generalizes well across distinct data distributions. All main-text experiments were systematically repeated on these datasets. Here are the anonymous link for additional experimental results: [TFIM](https://tinyurl.com/47mpp43c) and [XXZ-Heisenberg](https://tinyurl.com/3r598nmm).
>
> We consistently observe that margins strongly correlate with generalization gaps across multiple datasets and experimental conditions, significantly strengthening the relevance and robustness of our margin bounds for QNNs.
>
> ---
> ### 2) **Unified Theorem Incorporating Quantum Data Embedding**:
> With [Neural Quantum Embedding](https://tinyurl.com/4nyhdv5k), the quantum classifier for classical data is constructed through a two-step training procedure: first optimizing only the embedding circuit, and subsequently training the quantum neural network while keeping the embedding circuit fixed. Developing a fully unified theory would require separately deriving generalization bounds for the embedding optimization stage—which fundamentally differs from the classification task—and then integrating these results into a bound for overall classification performance. Such an analysis is inherently non-trivial and is currently beyond the scope of our work.
>
> Note that our existing margin-based bound implicitly captures the embedding circuit parameters, as these parameters directly influence the margin distribution of the trained model. Nevertheless, we agree that developing unified theoretical bounds explicitly incorporating embedding circuit properties is an interesting direction for future research.
>
> Moreover, many quantum machine learning applications involve quantum data directly, with no classical-to-quantum embedding required. In these cases, explicitly incorporating embedding circuit properties is not relevant.
>
> ---
> ### 3) **Margin and Generalization Gap Analysis in Figures**:
>
> We confirm that Figures 1 and 4 initially emphasized test accuracies for clarity. However, margin distributions also strongly correlate with generalization gaps in these experiments. We have updated these figures to include both test accuracy and generalization gaps on this [anonymous link](https://tinyurl.com/4xtxt2a7).
>
> As predicted by our theory, we observe in Figure 1 (QPR) that generalization gaps consistently decrease as margin distributions shift rightward. For Figure 4 (MNISTs), we see the same trend explicitly for the MNIST and Fashion-MNIST. For the KMNIST, we observe a slight discrepancy, where the gap marginally increases.
>
> This minor deviation can be explained by the significantly larger number of samples (~12,000) relative to system complexity (8-qubit QCNN). Our margin bound scales as $1/\sqrt{m}$, where $m$ is the sample size. Thus, when the number of samples greatly exceeds system complexity, generalization gaps become naturally small, and margin effects appear less pronounced. Indeed, the gaps observed for the MNIST-based dataset are substantially smaller compared to those with the QPR dataset, which includes only 20 samples following the experimental setup of [Caro et al.](https://tinyurl.com/mrsrrjk4) and [Gil-Fuster et al.](https://tinyurl.com/4kjnvb44). Therefore, this subtle deviation aligns with our theory rather than contradicting it. We will explicitly clarify this nuance and present both test accuracies and generalization gaps clearly in the revised manuscript.
>
> In addition, we would like to clarify that generalization gap analysis was already presented in the main manuscript through Figures 2 and 3. These figures present the relationship between margin-based metrics and generalization gap across various architectural choices and data corruption levels, supporting our theoretical claims. This is also why we originally placed Figure 6, which provides supplementary generalization gap analysis comparing parameter-based metrics with the lower quartile and mean of the margin for the QPR problem, in the appendix.
>
> Lastly, our work substantially broadens the experimental scope compared to prior state-of-the-art studies, such as [Caro et al.](https://tinyurl.com/mrsrrjk4) and [Gil-Fuster et al.](https://tinyurl.com/4kjnvb44). While these focus on a single task involving quantum data, our experiments span multiple QML architectures and a range of both classical and quantum datasets. This broader scope enables a more comprehensive assessment of the practical relevance of margin-based generalization theory and deepens our understanding of generalization in QML.

---

> > ### Comment · Reviewer_uadb · 2025-04-04
> >
> > Thank you for adding additional experiments; I am increasing my score to 3.

---

> > > ### Author Response · Authors · 2025-04-09
> > >
> > > We sincerely thank Reviewer uadb for your thoughtful evaluation and for recognizing our additional experiments. We greatly appreciate your constructive feedback.

---

### Official Review · Reviewer_tcJi · 2025-03-14

**Overall Recommendation:** 3

**Summary:**

This paper provides generalization error upper bounds for parameterized quantum neural networks using arguments from Bartlett et al. (2017) 's construction.

**Claims And Evidence:**

I find all the claims in the paper to be reasonable. What helps this work is that there is a long line of work in deriving generalization error bounds based on margins starting from Bartlett (1996) [cited in the paper] and Bartlett et al. (1998) [1]. This also constitutes a drawback of the paper, in my opinion - the theoretical contributions of this paper are a very straightforward generalization of existing techniques. I invite the authors to correct me on this point if I have missed any non-trivialities in A.1 that they feel should be highlighted as an important contribution.

[1]  Peter Bartlett. Yoav Freund. Wee Sun Lee. Robert E. Schapire. "Boosting the margin: a new explanation for the effectiveness of voting methods." Ann. Statist. 26 (5) 1651 - 1686, October 1998. https://doi.org/10.1214/aos/1024691352.

**Essential References Not Discussed:**

N/A

**Experimental Designs Or Analyses:**

No issues.

**Methods And Evaluation Criteria:**

I find the proposed evaluation criteria to be acceptable. However, since I am not an experimentalist, I will defer to the judgment of other reviewers.

**Other Comments Or Suggestions:**

See the above comments. I am willing to raise my score after the rebuttal period based on discussions with other reviewers on the novelty in the experimental section. As it stands, despite the paper being well written, its contributions are incremental (merely quantizing existing analysis does not constitute a good contribution in my opinion), I cannot justify recommending acceptance.

**Other Strengths And Weaknesses:**

This paper should only be accepted at a premier ML conference like ICML if the experimental section is worthy of being highlighted, since the theoretical contributions are incremental at best in my opinion. This is not meant as a targeted criticism of the author's efforts - my issue is that margin bounds have long since shown to be loose for generalization bounds (especially for overparameterized circuits).

---------------------------------------------
Post Rebuttal
---
I am still unconvinced on the second point. However I am raising my score to reflect that I no longer stand by the "incremental advance" part.

**Questions For Authors:**

See Claims section.

**Relation To Broader Scientific Literature:**

This paper makes a nice addition to the literature on the generalization bounds of parameterized QNNs.

**Theoretical Claims:**

I have gone over the theoretical parts of this paper in detail (and I am already familiar with the existing papers in the literature) and I don't see any obvious issues with the theoretical claims of this paper.

---

> ### Author Rebuttal · Authors · 2025-03-31
>
> We thank Reviewer tcJi for constructive feedback.
>
>
> We understand the concern that our theoretical contribution could appear incremental, given the established history of margin-based bounds in classical ML. However, our work is the first to systematically extend this theoretical framework to QNNs. This extension is neither straightforward nor trivial. The translation of classical techniques to the quantum setting requires addressing unique mathematical and physical challenges that arise from quantum information processing, and in doing so, our work makes the following nontrivial technical advances:
>
> 1. **Lipschitz Continuity of POVMs** (Sec.2): Unlike classical neural networks, QNNs employ POVMs. Handling the Lipschitz continuity of quantum measurements required specialized analysis unique to quantum states, involving spectral norms of POVM operators.
>
> 2. **Covering Number of Complex Unitary Matrices** (A.1): Classical margin bounds rely on covering numbers of real-valued neural networks. By contrast, QNNs require derivations for complex-valued unitary matrices constrained by quantum mechanics. This differentiates our theoretical approach from existing classical frameworks.
>
> 3. **Normalization properties of quantum states** (A.1): Quantum states possess normalization constraints that substantially simplify and alter complexity arguments. We exploit this property to derive tighter bounds, a simplification with no direct classical counterpart.
>
> 4. **Extension to mixed input states via vectorization** (A.1): Our margin bound is further generalized to mixed quantum states utilizing vectorization arguments. This step is fundamentally quantum-specific, with no classical counterpart.
>
> Notably, our work addresses key open problems raised by two recent publications in Nature Communications ([13:4919, 2022](https://www.nature.com/articles/s41467-022-32550-3) and [15:2277, 2024](https://www.nature.com/articles/s41467-024-45882-z)), which represent the state of the art in QML generalization theory. The first introduces a generalization bound based on Rademacher complexity, while the second demonstrates the vacuity of such uniform bounds through extensive randomization experiments, concluding that existing theoretical tools are insufficient to explain generalization in QML.
>
> Our work goes beyond these foundational studies by introducing a margin-based framework that more accurately predicts generalization behavior in QNNs. In addition, our analysis applies to arbitrary quantum states—pure or mixed—whereas the prior works provide theoretical guarantees only for pure states. Rather than being a routine extension of classical results, our contribution directly addresses the theoretical gap identified in the second Nature Communications paper and significantly advances the understanding of generalization in QML beyond what is provided in the first.
>
> Moreover, our work establishes a previously unexplored theoretical connection between margin bounds and quantum state discrimination. This provides not only conceptual insight but also practical guidance for optimizing quantum data embedding. Specifically, by directly linking margins to trace distances, our theory offers a principled route for improving generalization performance when applying QNNs to classical datasets. This perspective also enables us to systematically optimize the quantum feature maps introduced in [Nature 567, 209–212 (2019)](https://www.nature.com/articles/s41586-019-0980-2), which proposed using quantum-enhanced feature spaces for supervised learning. We experimentally demonstrate that our margin-based approach significantly improves classification accuracy compared to the method in the Nature paper (Fig. 4). Thus, our work not only advances the theoretical foundations of QML, but also delivers a concrete performance gain over one of its most recognized experimental baselines.
>
> Additionally, our experimental section provides novel and extensive evidence showing that margin-based metrics outperform standard parameter-count metrics across multiple challenging scenarios, including randomized labels, varying network depths and ansatz, and different embedding methods. In response to Reviewer 2’s suggestion, we have added additional experiments using the [Transverse Field Ising Model](https://tinyurl.com/47mpp43c) and [XXZ-Heisenberg Model](https://tinyurl.com/3r598nmm), further validating the robustness and practical relevance of our margin-based framework.
>
> We believe these clarifications demonstrate the depth and originality of our contributions, which go beyond a trivial adaptation of classical techniques. By addressing open questions in QML with quantum-specific analysis and expanded empirical evidence, we believe our work offers timely and meaningful progress. We are grateful for the reviewer’s thoughtful feedback and hope this response helps convey the full value and relevance of our submission.

---

> > ### Comment · Reviewer_tcJi · 2025-04-02
> >
> > Thank you for the detailed response. I am raising my score to 3 for now, pending further discussions in post-rebuttal period.

---

> > > ### Author Response · Authors · 2025-04-09
> > >
> > > We sincerely thank Reviewer tcJi for reconsidering your evaluation and raising your score. We appreciate your thoughtful engagement and constructive discussions.
> > >
> > > Regarding your remaining reservation about the “second point,” we are not entirely certain whether it refers to the technical treatment of covering numbers in the complex unitary setting or the theoretical looseness of margin-based generalization bounds. To ensure completeness, we address both interpretations below.
> > >
> > > ---
> > > ###  **1. On covering number of complex unitary matrices:**
> > > Firstly, the quantum measurement function $g:\mathbb{C}^N \mapsto \mathbb{R}^k$ is $2 \sqrt{\sum_i \vert\vert E_i \vert\vert_\sigma}$-Lipschitz, meaning:
> > > $\vert\vert g(u) - g(v) \vert\vert_2 \leq 2 \sqrt{\vert\vert E_i \vert\vert_\sigma} \vert\vert u - v \vert\vert_2.$
> > > Note that the latter norm $ \vert\vert\cdot\vert\vert_2 $ denotes the complex vector 2-norm. Consequently, when peeling off quantum measurements, and reducing it to a matrix covering, it becomes essential to consider matrix coverings with respect to complex 2-norms.
> > >
> > > To rigorously achieve this, several subtle quantum-specific modifications were introduced in Appendix A.1:
> > >
> > > - We applied Maurey's Sparsification Lemma by introducing a discrete set:
> > > $$
> > > V = \lbrace V_1, \dots, V_{4N^2} \rbrace = \lbrace gY e_i e_j^\mathrm{T} : g \in \lbrace+1, -1, +i, -i\rbrace,\ i \in [N],\ j \in [N]\rbrace.
> > > $$ This discrete set, including complex constants $g \in \lbrace+1, -1, +i, -i\rbrace $, is essential to properly cover the complex matrix space.
> > >
> > > - We introduced a complex-adapted norm:
> > > $\vert\vert B \vert\vert_* = \sum_{i,j} \left(|\text{Re}(B_{ij})| + |\text{Im}(B_{ij})|\right)$,
> > > which is then upper-bounded using the complex Hölder’s inequality.
> > >
> > > These quantum-specific adjustments lead to subtle yet necessary modifications in the covering number bound (introducing factors such as 2 from $ g \in \lbrace+1, -1, +i, -i\rbrace $ and $ \sqrt{2} $ from the norm adjustments). Although subtle and hidden inside big-O notation at the conclusion, these careful steps were necessary to rigorously generalize the classical covering number analysis into the complex domain required for quantum machine learning.
> > >
> > > ### **2. On the looseness of margin-based generaization bounds:**
> > > We would like to emphasize that our work does not claim tight worst-case theoretical guarantees. Instead, we focus on showing that margin distribution–based quantities correlate more strongly with generalization behavior than conventional parameter-based uniform bounds commonly used in QML.
> > >
> > > In this sense, our framework offers a practical and interpretable alternative that empirically outperforms other generalization metrics in diverse QML scenarios. Nevertheless, we agree that exploring tighter bounds, particularly tailored to quantum settings, remains a valuable direction for future theoretical work.
> > >
> > >
> > > ---
> > > We hope this clarification effectively addresses your concerns.

---

### Decision · Program_Chairs · 2025-05-01

**Decision:**

Accept (poster)

**Comment:**

This paper extends previous results in classical learning theory on margin-based generalization to the quantum setting. It provides new generalization bounds for quantum neural networks (also known as parameterized quantum circuits), contributing to the theoretical understanding of generalization in quantum machine learning (QML). It also provides an experimental analysis to support the theoretical results. All the reviewers agree that this is a solid paper with valuable theoretical fundings. Concerns were raised regarding the experimental evaluation. The authors in their rebuttal provided additional experimental results to address these concerns. While further evaluation on devices with a larger number of qubits would strengthen the experimental part, this paper is a nice contribution to the field of QML and is worthy of publication. I recommend that the authors take into account all the reviewer’s comments to enhance the experimental results and the presentation of the paper (e.g., clear relevant work section).